# T cells discriminate between groups C1 and C2 HLA-C

Malcolm J W Sim*[†], Zachary Stotz, Jinghua Lu, Paul Brennan, Eric O Long, Peter D Sun*

Laboratory of Immunogenetics, National Institute of Allergy and Infectious Diseases, National Institutes of Health, Rockville, United States

**Abstract** Dimorphic amino acids at positions 77 and 80 delineate HLA-C allotypes into two groups, C1 and C2, which associate with disease through interactions with C1 and C2-specific natural killer cell receptors. How the C1/C2 dimorphism affects T cell recognition is unknown. Using HLA-C allotypes that differ only by the C1/C2-defining residues, we found that KRAS-G12D neoantigen-specific T cell receptors (TCRs) discriminated between C1 and C2 presenting the same KRAS-G12D peptides. Structural and functional experiments, and immunopeptidomics analysis revealed that Ser77 in C1 and Asn77 in C2 influence amino acid preference near the peptide C-terminus (pΩ), including the pΩ-1 position, in which C1 favors small and C2 prefers large residues. This resulted in weaker TCR affinity for KRAS-G12D-bound C2-HLA-C despite conserved TCR contacts. Thus, the C1/C2 dimorphism on its own impacts peptide presentation and HLA-C-restricted T cell responses, with implications in disease, including adoptive T cell therapy targeting KRAS-G12D-induced cancers.

*For correspondence:
mjwsim@gmail.com (MJWS);
PSUN@niaid.nih.gov (PDS)

Present address: †Centre for Immuno-Oncology, Nuffield Department of Medicine, University of Oxford, Headington, United Kingdom

Competing interest: The authors declare that no competing interests exist.

## Editor's evaluation

In this study, the authors use structural, functional, and immunopeptidomics analysis to provide insights into how HLA-C C1/C2 dimorphism impacts T cell recognition. This knowledge is important in immunotherapies targeting HLA-C-specific T cells.

## Introduction

The classical class I human leukocyte antigens (HLA-I; HLA-A, HLA-B, and HLA-C) are the most polymorphic genes across human populations and are associated with a myriad of human diseases (*Dendrou et al., 2018*; *Robinson et al., 2017*). HLA-I molecules present short peptides on the cell surface of nucleated cells where they form ligands for receptors of the immune system including T cell receptors (TCRs) and natural killer (NK) cell receptors. HLA-I-bound peptides are collectively known as the immunopeptidome, which defines the subset of the proteome bound as peptides to a particular HLA-I allotype (*Istrail et al., 2004*). Immunopeptidomes can be highly diverse, consisting of hundreds to thousands of different sequences as they are shaped by polymorphic amino acid residues that form pockets (A to F) in the HLA-I peptide-binding groove (PBG) (*Di Marco et al., 2017*; *Madden, 1995*; *Sarkizova et al., 2020*). These pockets restrict the peptide repertoire to those peptides with certain amino acid side chains at particular positions, which form anchor residues that define allotype-specific motifs (*Di Marco et al., 2017*; *Falk et al., 1991*; *Sarkizova et al., 2020*). The 'anchor' residues are often position 2 (p2) or p3 (B and C pockets) and the C-terminus (pΩ; F pocket), while residues outside the anchors can vary extensively allowing immunopeptidome diversification (*Di Marco et al., 2017*; *Falk et al., 1991*; *Madden, 1995*; *Sarkizova et al., 2020*). HLA-I allotypes with similar PBGs bind similar peptides, but subtle amino acid differences can modify the side-chain

orientation impacting interactions with TCRs and NK cell receptors (*Illing et al., 2018*; *Illing et al., 2012*; *Saunders et al., 2020*; *Stewart-Jones et al., 2012*; *Tynan et al., 2005*). Understanding how HLA-I polymorphism impacts the immunopeptidome and interactions with immunoreceptors is critical for constructing molecular mechanisms that underpin disease risk associated with specific HLA-I alleles (*Dendrou et al., 2018*).

HLA-C is thought to play a lesser role in T cell responses compared to that of HLA-A and HLA-B due to its lower cell surface expression level (*Apps et al., 2015*; *McCutcheon et al., 1995*). However, HLA-C-restricted T cells are implicated in several disease settings. In HIV infection, HLA-C allotypes with a high expression were associated with increased T cell responses and slower progression to AIDS, while higher HLA-C expression was also linked with susceptibility to Crohn's disease (*Apps et al., 2013*). In psoriasis, the strongest genetic association is with *HLA-C\*06:02*, thought to be mediated by HLA-C\*06:02-restricted T cells (*Chen and Tsai, 2018*; *Lande et al., 2014*). Furthermore, recent efforts to define the specificities of tumor-infiltrating lymphocytes (TIL) readily identified HLA-C-restricted T cells (*Levin et al., 2021*; *Lu et al., 2014*; *Murata et al., 2020*; *Tran et al., 2015*; *Tran et al., 2016*). Most notably, multiple HLA-C\*08:02-restricted TCRs specific for the oncogenic hotspot mutation KRAS-G12D were identified in several cancer patients (*Tran et al., 2015*; *Tran et al., 2016*). Adoptive transfer of expanded KRAS-G12D-specific TILs successfully treated a patient with metastatic colorectal cancer, leading to complete regression of all but one metastatic lesion (*Tran et al., 2016*). The remaining lesion lost *HLA-C\*08:02* from its genome demonstrating conclusively that HLA-C-restricted T cells can mediate effective antitumor responses (*Tran et al., 2016*). Despite the clinical relevance of HLA-C-restricted T cells, few studies have examined TCR-HLA-C interactions with a molecular and structural focus. Indeed, our recent study on HLA-C\*08:02-restricted KRAS-G12D-specific TCRs was the first to present crystal structures of any TCR in complex with HLA-C (*Sim et al., 2020*).

*HLA-C* is the most recently evolved classical HLA-I gene, present in only humans and other apes, and has several unique features (*Adams and Parham, 2001*). HLA-C, not HLA-A or HLA-B, is expressed on fetal extravillous trophoblast cells, making it the most polymorphic molecule expressed at the maternal–fetal interface (*King et al., 2000*). In addition, all HLA-C allotypes serve as ligands for the killer-cell immunoglobulin-like receptors (KIR), whereas only a subset of HLA-A and HLA-B allotypes bind KIRs (*Hilton and Parham, 2017*; *Moesta et al., 2008*). The KIR are a family of activating and inhibitory receptors expressed primarily on NK cells (*Hilton and Parham, 2017*; *Saunders et al., 2015*). HLA-C allotypes form two groups, C1 and C2, based on two dimorphic amino acid residues at positions 77 and 80 (*Biassoni et al., 1995*; *Parham et al., 2012*). C1 HLA-C allotypes carry Ser77 and Asn80 and are ligands for the inhibitory receptors KIR2DL2/3. The inhibitory receptor KIR2DL1 binds C2 HLA-C allotypes, which carry Asn77 and Lys80. The KIR discriminate C1 and C2 HLA-C largely via position 44 in the KIR (*Winter and Long, 1997*) and direct interactions with HLA-C position 80 (*Boyington et al., 2000*; *Boyington and Sun, 2002*; *Fan et al., 2001*). Genetic association studies link C1/C2 status and the presence or absence of specific KIR-HLA combinations with risk of multiple human diseases, including cancer, infectious diseases, autoimmunity, and disorders of pregnancy (*Hiby et al., 2004*; *Khakoo et al., 2004*; *Kulkarni et al., 2008*; *Littera et al., 2016*; *Parham and Moffett, 2013*; *Rajagopalan and Long, 2005*; *Venstrom et al., 2012*). Whether the C1/C2 dimorphism impacts HLA-C interactions with other immunoreceptors is unknown.

Given the clinical relevance of HLA-C-restricted T cells and that all HLA-C allotypes are either C1 or C2, it is important to ask whether this dimorphism impacts T cell recognition. It was hitherto assumed that the C1/C2 dimorphism would not directly affect T cell recognition as it lies outside the general TCR footprint. To answer this question, we studied two HLA-C\*08:02 (C1)-restricted TCRs specific for different KRAS-G12D neoantigens. TCR9a is specific for G12D-9mer (GA**D**GVGKSA), while TCR10 is specific for G12D-10mer (GA**D**GVGKSAL) (*Sim et al., 2020*; *Tran et al., 2016*). TCR9a and TCR10 do not share TCR V genes and recognize their peptide antigens in different ways (*Sim et al., 2020*; *Tran et al., 2016*). We compared TCR9a and TCR10 recognition of HLA-C\*08:02 (C1) with that of HLA-C\*05:01, a C2 allotype that is identical in sequence to HLA-C\*08:02 apart from the C1/C2 defining residues at positions 77 and 80 (*Sim et al., 2017*). In a previous study, we showed that 28 different 'self' peptides eluted from HLA-C\*05:01 bound to both HLA-C\*05:01 (C2) and HLA-C\*08:02 (C1), suggesting a minimal impact of the C1/C2 dimorphism on peptide binding to HLA-C (*Sim et al., 2017*). Here, we found that T cells discriminate between C1 and C2 HLA-C due to differences in

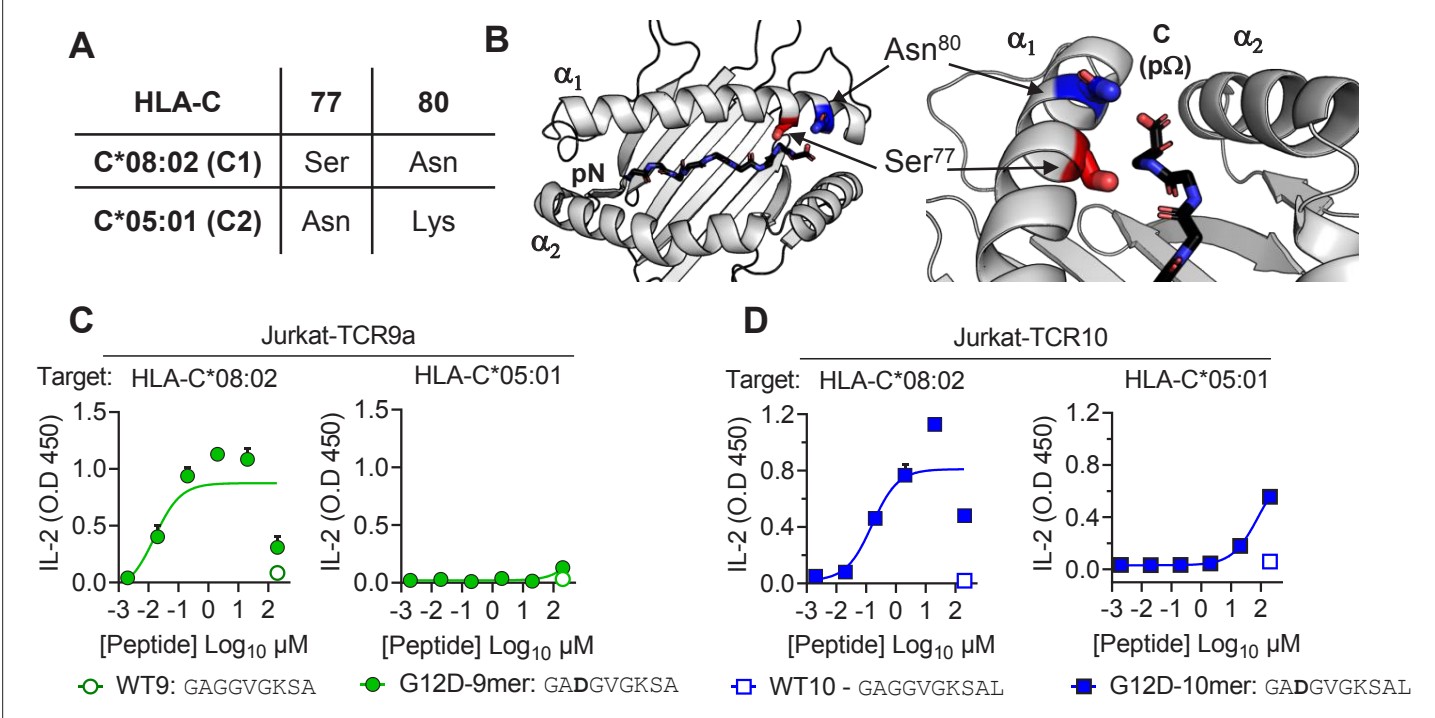

**Figure 1.** HLA-C C1/C2 dimorphism impacts T cell recognition of KRAS-G12D neoantigen. (**A**) Sequence of C1/C2 dimorphism in HLA-C*08:02 and HLA-C*05:01 at positions 77 and 80. All other residues are identical. (**B**) Location of C1/C2 dimorphism on the structure of HLA-C*08:02, close to peptide C-terminus (PDB:6ULI). (**C, D**) Stimulation of TCR9a+ Jurkat cells (**C**) and TCR10+ Jurkat cells (**D**) by 221C*08:02 (left) or 221C*05:01 (right) preloaded with KRAS WT or G12D peptides at indicated concentrations. Means and standard errors of IL-2 concentration in culture supernatant measured by ELISA are shown from two independent biological replicates. Source data available in *Figure 1—source data 1*.

The online version of this article includes the following source data and figure supplement(s) for figure 1:

**Source data 1.** ELISA readings for *Figure 1C, D*.

**Figure supplement 1.** Impact of C1/C2 dimorphism in T cell recognition of HLA-C.

---

peptide binding to HLA-C and in peptide presentation to TCR, which revealed unsuspected features of the C1/C2 dimorphism governing interactions of HLA-C with immunoreceptors. Further, our data suggest that HLA-C*05:01-positive individuals may be eligible for immunotherapies targeting KRAS-G12D-10mer.

## Results

### Impact of the C1/C2 dimorphism to T cell recognition of KRAS neoantigens

HLA-C*05:01 (C*05) and HLA-C*08:02 (C*08) differ only by the C1/C2 dimorphism defined by positions 77 and 80, which are located on the α1 helix, near the peptide C-terminal amino acid (pΩ) (*Figure 1A and B*). We expressed C*08 and C*05 to similar levels in 721.221 cells, which are otherwise deficient in classical HLA-I, HLA-A, -B and -C (*Figure 1—figure supplement 1A*). These cells displayed canonical dimorphic recognition by KIR, with KIR2DL2/3 demonstrating strong binding to C*08, and KIR2DL1 demonstrating exclusive binding to C*05 (*Figure 1—figure supplement 1B*). We next used these cells as targets in functional experiments with Jurkat-T cells expressing C*08-restricted KRAS-G12D-specific TCRs, TCR9a, or TCR10. As expected, both the 9mer and 10mer G12D but not their WT KRAS peptides activated their respective TCR9a and TCR10 expressing T cells in the presence of C*08 (*Figure 1C and D*; *Sim et al., 2020*). Notably, we observed no activation of TCR9a and only weak activation of TCR10 when the 9mer and 10mer G12D KRAS peptides were presented by C*05, respectively (*Figure 1C and D*). To confirm this was not due to inefficient peptide loading, we utilized C*08 and C*05-expressing cell lines deficient in transporter associated with antigen presentation (TAP)

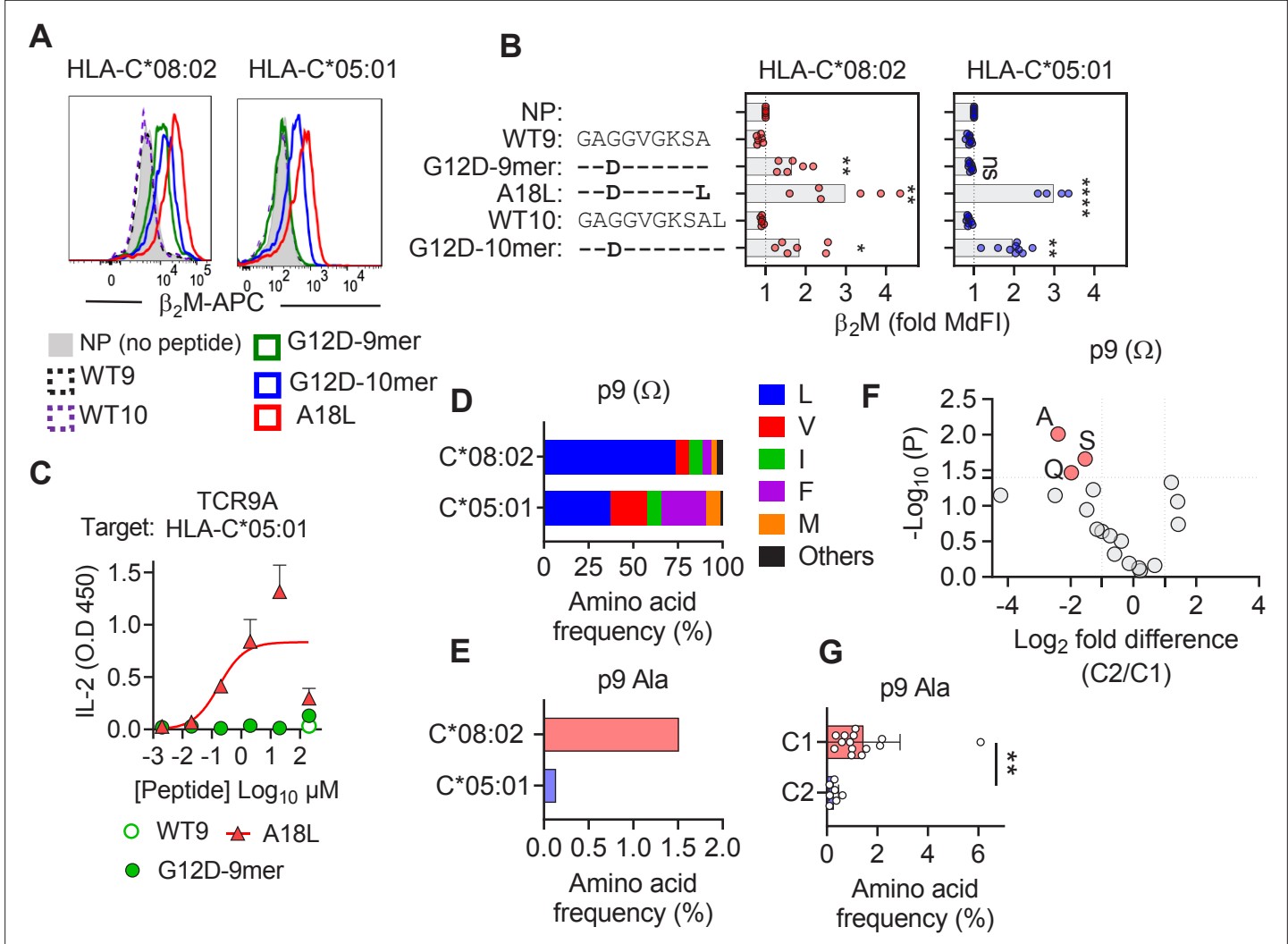

**Figure 2.** C1 but not C2 HLA-C allotypes present peptides with C-terminal (pΩ) Ala. (**A**) Stabilization of HLA-C on TAP-deficient 221 cells expressing HLA-C*08:02 or HLA-C*05:01. (**B**) Data from (**A**) shown as fold median fluorescent intensity (MdFI) relative to no peptide (NP) from a minimum of four independent experiments. (**C**) Stimulation of TCR9a⁺ Jurkat cells by 221C*05:01 cells preloaded with KRAS peptides at indicated concentrations. Means and standard errors of IL-2 concentration in culture supernatant measured by ELISA from two biological replicates are shown. (**D**) Frequency of indicated residues at the C-terminus (pΩ) in peptides eluted from HLA-C*08:02 or HLA-C*05:01. (**E**) Frequency of pΩ Ala in peptides eluted from HLA-C*08:02 or HLA-C*05:01. (**F**) Volcano plot displaying pΩ amino acid frequency from 21 HLA-C allotypes. The C2/C1 ratio is shown for the average frequency of each amino acid. (**G**) The frequency of pΩ Ala from 21 HLA-C allotypes by C1/C2 status. Statistical significance was assessed by unpaired *t*-test with Welsh's correction, *p<0.05, **p<0.001, ****p<0.0001. Source data available in *Figure 2—source data 1*. Peptide sequences and p9 frequency analysis are available in *Figure 2—source data 2*.

The online version of this article includes the following source data and figure supplement(s) for figure 2:

**Source data 1.** Normalized MdFI values for Figure 2B, ELISA readings for *Figure 2C* and amino acids frequencies for *Figure 2D-G*.

**Source data 2.** HLA-C Peptide sequences used for analysis.

**Figure supplement 1.** Impact of peptide length on frequency of C-terminal (pΩ) Ala in peptides eluted from C*08:02 and C*05:01.

(*Figure 1—figure supplement 1C*). These experiments yielded similar results using a different functional readout (expression of CD69), as G12D-9mer loaded on C*05 did not activate Jurkat-TCR9a cells, while G12D-10mer loaded on C*05 did stimulate Jurkat-TCR10 cells but required higher peptide concentrations than when loaded on C*08 (*Figure 1—figure supplement 1C*). Thus, the HLA-C1/C2 dimorphism impacts T cell recognition of KRAS-G12D neoantigen even though neither residue 77 nor 80 is in contact with the TCR. We next explored if the differences in T cell recognition of C*08 and C*05 were due to differences in the binding of peptide (*Figure 2*) or TCR (*Figure 3*) to HLA-C.

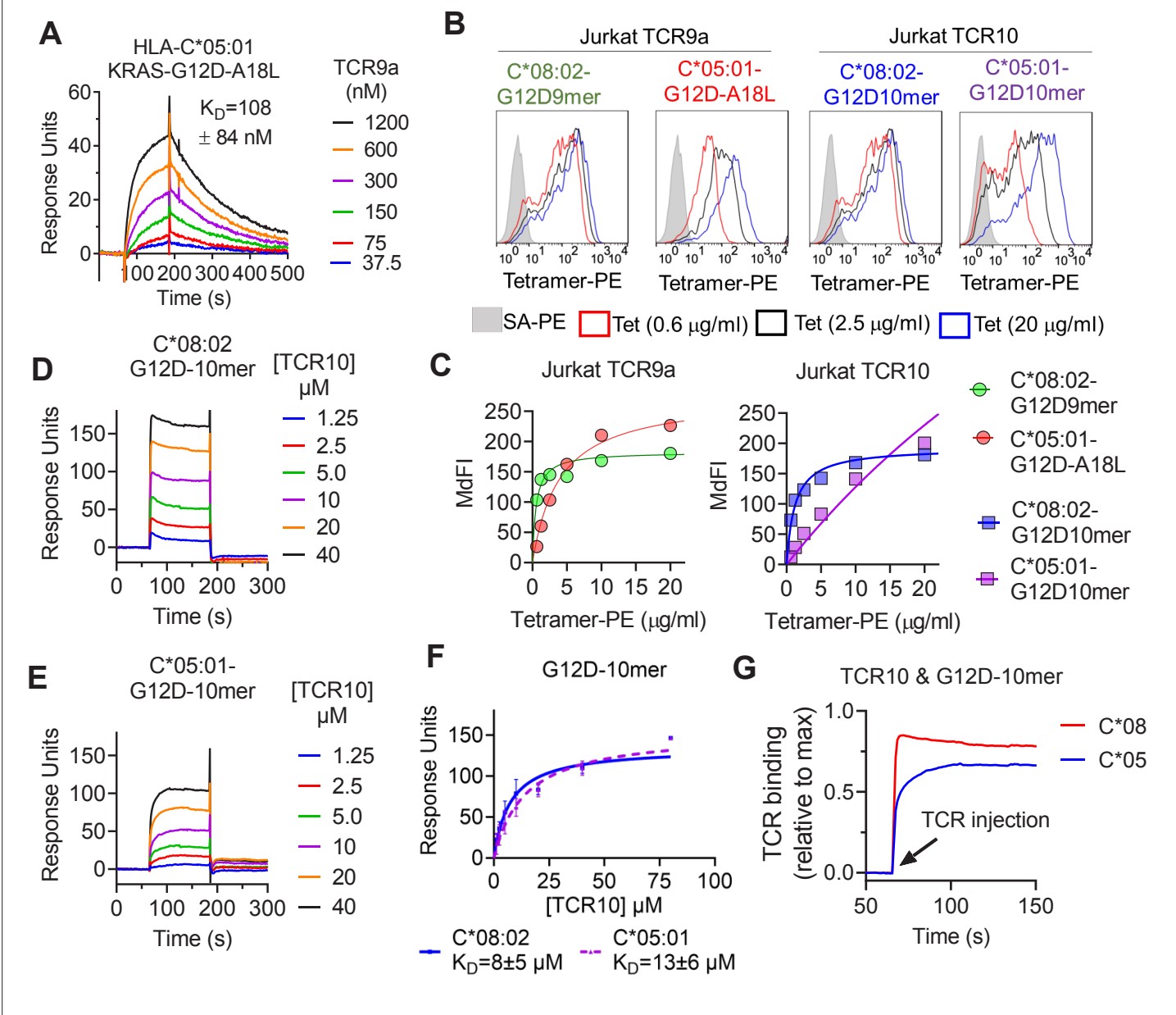

**Figure 3.** T cell receptor (TCR) binding is weaker to C2 HLA-C. (**A**) Binding of TCR9a to captured HLA-C*05:01-KRAS-G12D-A18L determined by surface plasmon resonance (SPR). Dissociation constant was determined by kinetic curve fitting of 12 curves from two independent experiments. (**B, C**) Binding of HLA-C*08:02 or HLA-C*05:01 tetramers to Jurkat T-cells expressing TCR9a or TCR10 at indicated tetramer concentrations. HLA-C was refolded with indicated peptides. (**C**) Summary of (**B**) from two independent experiments displayed as average (mean) and standard error of median fluorescence intensity (MdFI). (**D, E**) Binding of TCR10 to captured HLA-C*08:02-G12D-10mer and HLA-C*05:01-G12D-10mer by SPR. One experiment of four is shown. (**F**) Mean, standard deviation, and nonlinear curve fitting of TCR10 binding to G12D-10mer bound to C*08:02 and C*05:01. Dissociation constants were derived from four independent binding experiments with twofold serial dilutions of TCR10 starting at 10 µM (two experiments), 40 µM (one experiment), and 80 µM (one experiment). (**G**) Association binding of TCR10 with G12D-10mer presented by C*08:02 or C*05:01. SA, streptavidin; Tet, tetramer. Source data available in *Figure 3—source data 1*.

The online version of this article includes the following source data and figure supplement(s) for figure 3:

**Source data 1.** Raw data for TCR response curves in *Figure 3A, D, E and G*.

**Figure supplement 1.** C*08 is a better ligand for TCR9a than C*05.

## KRAS peptide C-terminal residue distinguishes C1 and C2 HLA-C allotypes

To characterize KRAS-G12D peptide binding to C*08 and C*05, we carried out peptide loading assays with TAP-deficient cells expressing C*08 or C*05. Similar to C*08, C*05 was not stabilized by WT-KRAS peptides likely due to the absence of Asp/Glu at p3, a critical anchor for C*08 and C*05 (*Figure 2A and B*). However, G12D-9mer only stabilized C*08 and not C*05, while G12D-10mer stabilized both allotypes (*Figure 2A and B*). An unusual feature of the G12D-9mer was pΩ Ala as its short side chain does not fill the pΩ pocket (F pocket), in contrast to G12D-10mer with pΩ Leu (*Sim et al., 2020*). We previously demonstrated that substitution of pΩ Ala to Leu in G12D-9mer (A18L-9mer: [10]GA**D**GVGKS**L**) improved binding to HLA-C*08:02, and T cell recognition by Jurkat-TCR9a (*Sim et al., 2020*). Similarly, A18L-9mer bound C*05 well (*Figure 2A and B*) and stimulated Jurkat-TCR9a cells (*Figure 2C*, *Figure 1—figure supplement 1C*). Thus, the inability of G12D-9mer to bind C*05 accounts for its lack of activation of Jurkat-TCR9a cells. Further, these data suggest that the C1/C2 dimorphism selectively impacts peptide binding to HLA-C in a sequence-dependent manner.

To explore if pΩ Ala is a general feature of peptides that bind C1 allotypes (like C*08) and not C2 allotypes (like C*05), we examined C-terminal amino acid usage in the immunopeptidomes of HLA-C allotypes using two publicly available datasets (see Materials and methods) (*Di Marco et al., 2017*; *Sarkizova et al., 2020*). We initially analyzed both datasets independently and observed similar amino acid frequencies, thus we present a combined analysis of a merged dataset with duplicate sequences removed. However, we cannot rule out that differences in cell line origin, data acquisition, false discovery rates, or analytical pipelines may have influenced our analysis. In addition, our analysis was limited to that of amino acid frequency and did not incorporate peptide abundance. We first compared the immunopeptidomes of C*08 and C*05 as the residues that form the pΩ binding pocket of C*08 and C*05 are identical and accordingly the five most abundant pΩ residues (Leu, Val, Ile, Phe, and Met) were the same in both allotypes, accounting for 97% of C*08% and 98% of C*05 sequences (*Figure 2D*). Leucine was the most common pΩ in both allotypes but had a higher frequency in C*08 peptides than C*05 (C*08 = 73%, C*05 = 37%) (*Figure 2D*). The significance of this difference is unknown; however, peptides with pΩ Leu, Val, Ile, Phe, and Met bind well to both C*08 and C*05 (*Sim et al., 2017*). Among 9mer peptides, the frequency of Ala at pΩ position was 1.5% (30 out of 1986) in C*08, significantly higher (p=0.0016 Fisher's exact test) than that of 0.13% (1 out of 732) in C*05-bound peptides (*Figure 2E*). Similar results were observed for 10mers, where pΩ Ala was more common among C*08 peptides (5/349, 1.4%) than C*05 peptides (0/152, 0%) (*Figure 2—figure supplement 1*). We next analyzed the pΩ amino acid frequency amongst 9mer sequences eluted from 14 C1 allotypes and 7 C2 allotypes and looked for amino acids with the greatest difference in frequency between C1 and C2 eluted peptides. In concordance with the comparison of C*05 and C*08 immunopeptidomes, pΩ Ala was the one most enriched in C1 allotypes, with a frequency of 1.45% ± 1.4% in C1 allotypes compared to 0.27% ± 0.19% in C2 allotypes (*Figure 2F and G*). In addition, Gln and Ser were significantly enriched at pΩ of C1 allotypes, but at lower frequencies than Ala. Thus, C1 allotypes bind peptides with pΩ Ala to a greater extent than C2 allotypes, consistent with the ability of G12D-9mer to bind C*08 and not C*05.

## Impact of C1/C2 dimorphism on TCR binding to HLA-C

We next explored the impact of the C1/C2 dimorphism on TCR binding to HLA-C using recombinant HLA-C refolded with individual peptides. The solution-binding affinities of the two neoantigen-specific TCRs, TCR9a and TCR10, to their cognate peptides presented by C*08 or C*05 were determined by surface plasmon resonance (SPR) using recombinant TCRs as analytes. As G12D-9mer with pΩ Ala does not bind C*05 (*Figure 2A*), we chose the Leu anchor variant A18L-9mer as the peptide for C*05 to ensure HLA-C stability. While TCR9a bound to the C1 allotype HLA-C*08 in the presence of G12D-9mer peptide with 16 nM affinity (*Sim et al., 2020*), it displayed a binding affinity of 108 nM for the C2 allotype HLA-C*05 in the presence of Leu anchor variant of G12D-9mer, an approximately sevenfold affinity reduction compared to the C1 allotype (*Figure 3A*, *Figure 3—figure supplement 1F*). Consistent with their solution-binding affinities, the C*08-G12D-9mer tetramers bound TCR9a-expressing Jurkat cells with lower EC$_{50}$ than C*05-A18L-9mer tetramers (*Figure 3B and C*). This reduced binding affinity and avidity may explain the reduced sensitivity of Jurkat-TCR9a cells for A18L-9mer loaded on C*08 compared to C*08 (*Figure 3—figure supplement 1B*). In solution, TCR10 bound both the C1

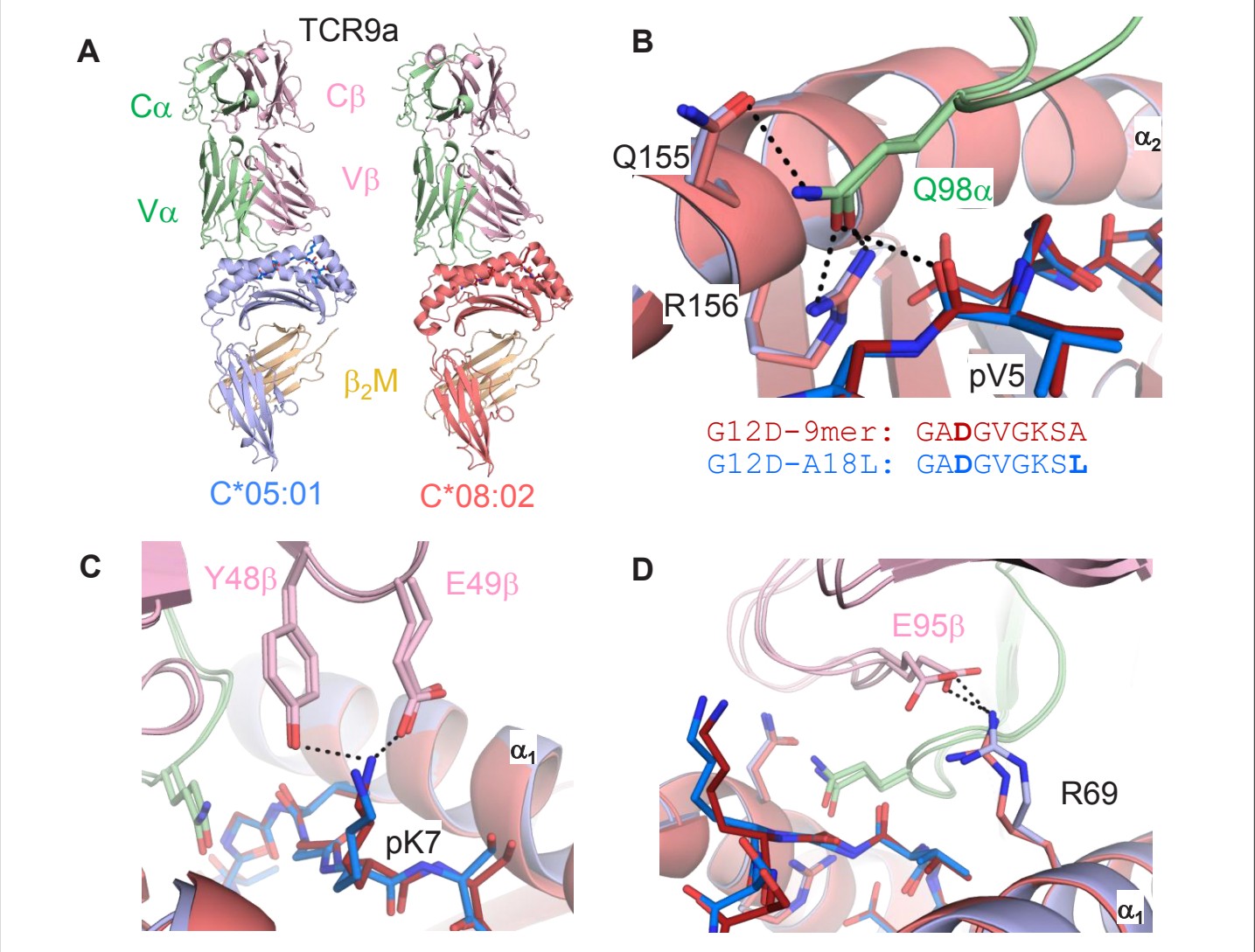

**Figure 4.** Minimal impact of C1/C2 dimorphism on TCR:HLA-C complex structure. (**A**) Side view of TCR9a in complex with HLA-C*05:01-G12D-A18L (left) and HLA-C*08:02-G12D-9mer (right; PDB:6ULR). (**B–D**) Interactions of TCR9a with HLA-C*05:01-A18L (blue) and HLA-C*08:02-G12D-9mer (red). TCR9a α chain, green; β chain, pink; β2-microglobulin (β$_2$M), gold.

The online version of this article includes the following figure supplement(s) for figure 4:

**Figure supplement 1.** Modeling TCR10 binding to C*05:01-G12D-10mer.

and C2 allotypes with similar μM affinities (**Figure 3D–F**), though slightly weaker affinity was observed with C*05. By steady-state kinetics, TCR10 binding was less than twofold weaker to C*05:01-G12D-10mer ($K_D$ = 13 ± 6 μM) than C*08:02-G12D-10mer ($K_D$ = 8 ± 5 μM) (**Figure 3D and E**). However, by kinetic analysis of the binding curves (**Figure 3G**), TCR10 had almost 20-fold slower on rate with C*05:01-G12D-10mer ($K_{on}$ = 6.2 ± 1.2 × 10$^3$ M$^{-1}$ s$^{-1}$) than with C*08:02-G12D-10mer ($K_{on}$ = 1.2 ± 0.9 × 10$^5$ M$^{-1}$ s$^{-1}$). Consistently, C*08-G12D-10mer exhibited better binding to TCR10-expressing Jurkat T cells than C*05-G12D-10mer, which failed to reach binding saturation at the highest concentration of tetramer tested (40 μg/ml) (**Figure 3B and C**). Thus, both TCR9a and TCR10 exhibited better binding to the C1 allotype than C2.

Neither TCR9a nor TCR10 make direct contacts to HLA-C positions Ser 77 and Asn 80 (**Sim et al., 2020**). To understand the impact of C2 residues Asn 77 and Lys 80 on T cell recognition, we determined the crystal structure of TCR9a in complex with C*05-A18L-9mer to a high resolution of 1.9 Å (**Supplementary file 1**). TCR9a adopted the same binding mode when in complex with C*05, as in the C*08 complex structure (**Figure 4A**). TCR9a recognition of C*05-A18L-9mer was via CDR3α Q98 that

forms hydrogen (h) bonds with Arg 156 and Gln 155 of HLA-C and the peptide backbone of peptide p5 Val (*Figure 4B*). TCR9a CDR2β Tyr48 and Glu49 coordinated peptide p7 Lys (*Figure 4C*), while CDR3β of TCR9a E95 formed a salt bridge with C*05 Arg 69 (*Figure 4D*). All these contacts observed between TCR9a and the C2 HLA-C*05 are conserved in the structure of TCR9a complexed with C*08 (*Supplementary file 2*). Like the C1 HLA-C*08, the dimorphic residues 77 and 80 on the C2 HLA-C*05 also do not interact with TCR9a. The closest TCR residue to HLA-C position 77 or 80 is Glu 49 of the TCRβ chain that interacts with Lys p7 of the peptide but is greater than 7 Å away from HLA-C position 80. Docking TCR10 onto a structure of C*05:01-G12D-10mer using our TCR10-C*08:02-G12D-10mer complex revealed no obvious impact on TCR contacts (*Figure 4—figure supplement 1*; *Bai et al., 2021*; *Sim et al., 2020*), suggesting the dimorphic positions on C*05 do not directly contact TCR.

## The C1/C2 dimorphism affects pΩ-1 backbone position and side-chain preference

The lack of direct contacts between TCR and the HLA-C1/C2 dimorphic residues raised the question if the C1/C2 dimorphism indirectly affected TCR binding through peptide conformation. The position 77 side chain is orientated directly into the PBG, while position 80 sits atop the α1-helix (*Figure 1B*). Position 77 is Ser in C*08 (C1) and Asn in C*05 (C2) and both form an h-bond of the same distance with the amide of the terminal peptide bond between pΩ and p8 (pΩ-1) (*Figure 5A*). Comparing the structures of C*05 and C*08, the HLA-C $\alpha_1$-helices were displaced relative to each other and peptide p8 was higher out of the PBG in C*05 compared to C*08 (*Figure 5B–D*). This displacement was measured by comparing the distance between the Cα of peptide p8 (Ser) with that of the Cβ of HLA-C position 77 (the first carbon of the position 77 side chain). This distance was 4.7 Å and 5.7 Å in the C*08 and C*05 structures, respectively (*Figure 5B and C*). We next examined the same distance in 10 additional published HLA-C structures, 4 C1 (HLA-C*03:04, PDB:1EFX, HLA-C*07:02, PDB:5VGE, HLA-C*08:01, PDB:4NT6, HLA-C*08:02-G12D-10mer, PDB: 6ULK) and 6 C2 (HLA-C*04:01, PDB:1QQD, HLA-C*05:01-G12D-10mer, PDB: 6JTO, HLA-C*05:01, PDB: 5VGD, HLA-C*06:02, PDB:5W6A, 5W69, 5W67). We observed that the difference between the Cα of peptide p8 and HLA-C 77 Cβ was maintained between C1 and C2 HLA-C structures (C1 = 4.6 ± 0.11 Å, C2 = 5.7 ± 0.22 Å) (*Figure 5E*). In contrast, a conserved contact with the peptide N terminus and Tyr171 Oγ was the same between C1 and C2 HLA-C structures (2.70 Å vs. 2.76 Å) (*Figure 5F and G*). A consequence of this displacement is the proximity of the p8 side chain to the HLA-C $\alpha_1$ helix (*Figure 5H*). We compared the distance between the p8 side chain and HLA-C Val 76 in the 10 HLA-C structures. In all five C1 allotype structures, the p8 side chain was within van der Waals range of Val 76 (3.3–4.0 Å). In contrast, the p8 side chain was out of contact range with Val 76 in all but one C2 HLA-C structures (*Figure 5H and I*). This appears to be due in part to the torsion angle of the terminal peptide bond that orientates p8 towards the $\alpha_1$ helix in C1 allotypes and out of the groove in C2 allotypes (*Figure 5J*). Thus, the C1/C2 dimorphism impacts the position and orientation of the peptide at pΩ-1, suggesting that position 77 is the major determinant of the difference between C*05 and C*08. To interrogate the role of positions 77 and 80 individually, each one was substituted in C*05 with the corresponding residue in C*08 (*Figure 5—figure supplement 1*). Cells expressing C*05, C*05 Asn77Ser (N77S), C*05 Lys80Asn (K80N), and C*08 loaded with G12D-9mer or G12D-10mer were used as targets for Jurkat-TCR9a and Jurkat-TCR10 cells, respectively. The lack of stimulation of Jurkat-TCR9a by C*05 was minimally impacted by C*05-K80N, while C*05-N77S recovered Jurkat-TCR9a responses, though not completely (*Figure 5K*). In contrast, stimulation of Jurkat-TCR10 cells by HLA-C*05:01 was not improved by single substitutions at positions 77 and 80, suggesting a synergistic effect of positions 77 and 80 on recognition of HLA-C*08:02 by TCR10.

We next explored if the shift in pΩ-1 position is associated with changes in the HLA-C immunopeptidome other than those at pΩ (*Figure 2*). Analyzing 9mer sequences unique to C*08 and C*05, we compared the amino acid frequency at each position (1–9) by Pearson correlation. Strong correlations ($r > 0.9$) were observed at positions 1–3, while the pΩ was slightly weaker ($r = 0.8$), reflecting the small differences in pΩ usage (*Figure 6A*, *Figure 6—figure supplement 1A*). In contrast, there was a weak correlation ($r = 0.46$) at p7 and no correlation in amino acid usage ($r = –0.01$) at p8 (pΩ-1) (*Figure 6A*). At pΩ-1, Ala and Ser made up over 30% of sequences unique to C*08, but only 5% of sequences unique to C*05. In contrast, Leu, Lys, and Arg were found in approximately 40% of sequences unique to C*05, but only 7% of sequences unique to C*08 (*Figure 6A*). We observed similar findings for p9

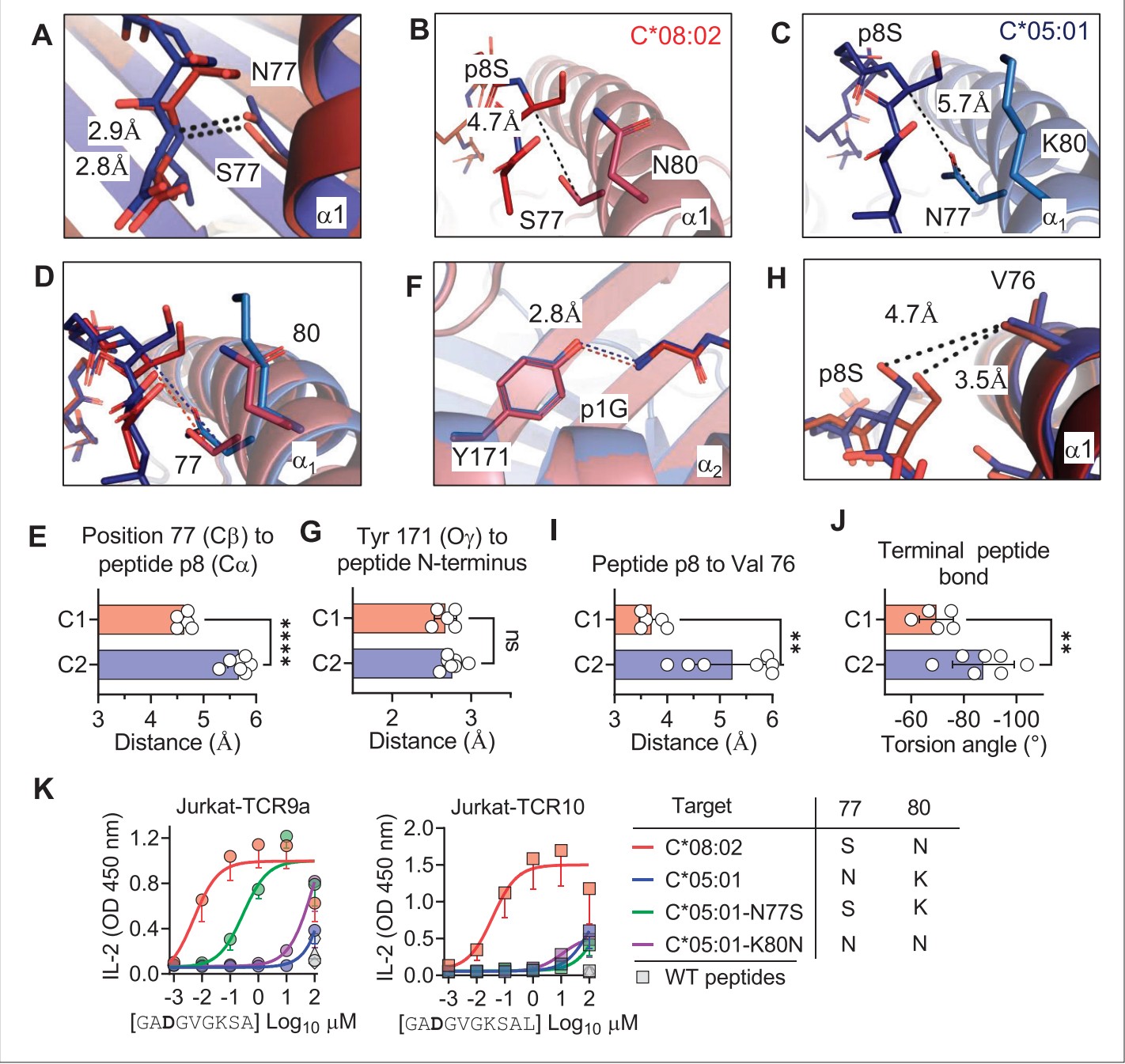

**Figure 5.** C1/C2 dimorphism determines the distance between peptide p8 and HLA-C. (**A**) Hydrogen bond between HLA-C position 77 and amide in terminal peptide bond in C*08:02-G12D-9mer and C*05:01-A18L-9mer structures. (**B–D**) Distance between HLA-C position 77 Cβ and Cα of peptide p8 Ser of HLA-C*08:02-G12D-9mer (**B**), HLA-C*05:01-G12D-A18L (**C**), and both overlaid (**D**). (**E**) Distances between HLA-C position 77 Cβ and Cα of peptide pΩ-1 in 12 HLA-C crystal structures. (**F**) Distance between Tyr 171 Oγ and peptide N-terminus of C*08:02 and C*05:01. (**G**) Distance between Tyr 171 Oγ and peptide N-terminus in 12 HLA-C crystal structures. (**H**) Distance between peptide p8 side chain and HLA-C position 77 Val side chain in structures of C*08:02 and C*05:01. (**I**) Distance between peptide pΩ-1 side chain and HLA-C position 76 Val side chain in 12 HLA-C structures. (**J**) Torsion angle of terminal peptide bond in 12 HLA-C structures. (**K**) Stimulation of Jurkat-TCR9a+ and Jurkat-TCR10+ by 221 cells expressing HLA-C*08:02, HLA-C*05:01, HLA-C*05:01-N77S, or HLA-C*05:01-K80N preloaded with G12D-9mer (left) or G12D-10mer (right). Means and standard errors of IL-2 concentration in culture supernatant measured by ELISA from three biological replicates are shown. Significance in (**E**), (**G**), (**I**), and (**J**) was measured using an unpaired *t*-test with Welch's correction, **p<0.01, ****p<0.0001. Source data available in *Figure 5—source data 1*.

The online version of this article includes the following source data and figure supplement(s) for figure 5:

**Source data 1.** Raw distances for *Figure 5E, G, I*.

**Figure supplement 1.** Expression HLA-C in 721.221 cells with position 77 and 80 substitutions.

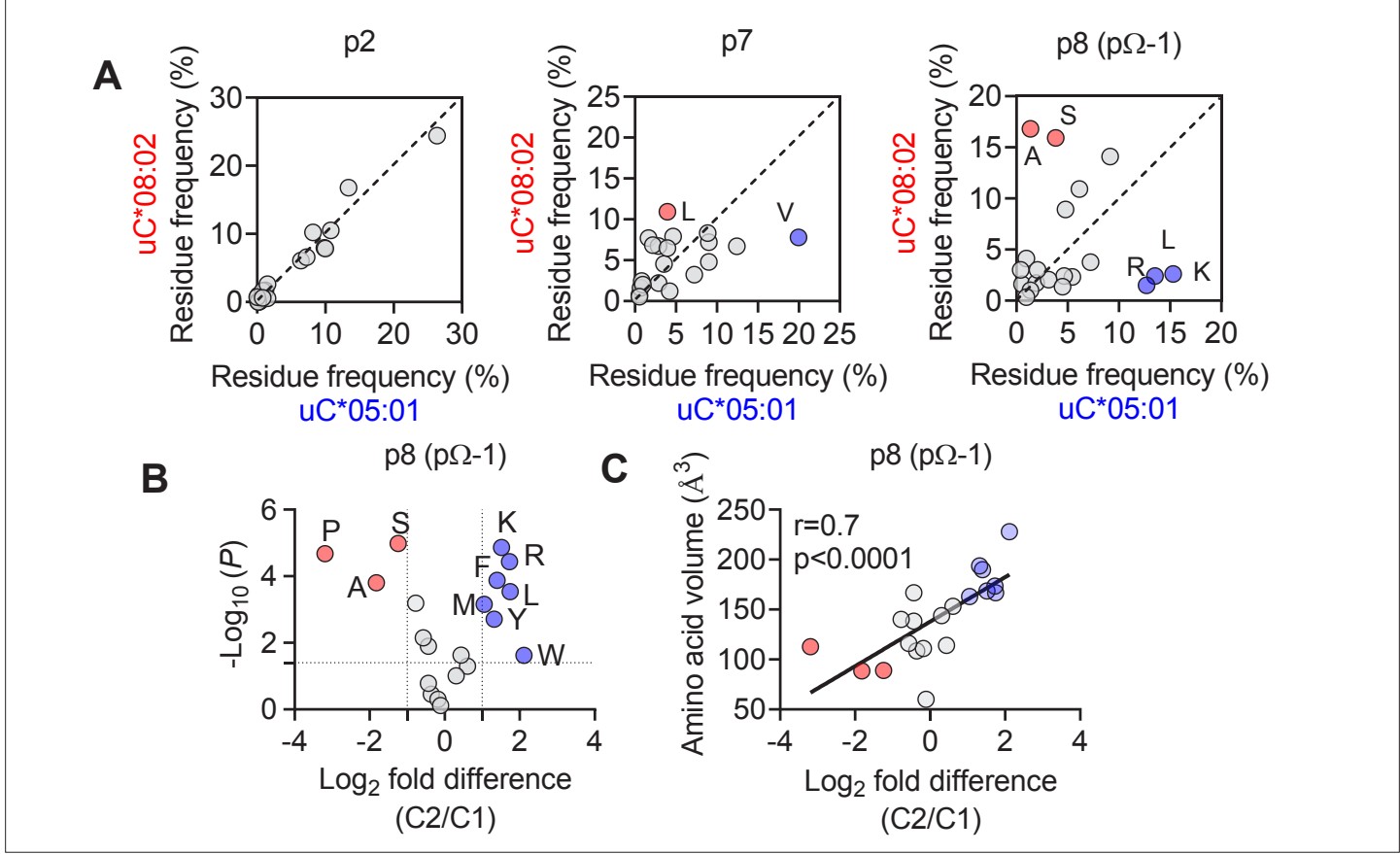

**Figure 6.** The C1/C2 dimorphism selects for side-chain size at pΩ-1. (**A**) Correlation of amino acid frequency at p2, p7, and p8 (pΩ-1) of 9mer peptides unique to HLA-C*08:02 (uC*08:02) and unique to HLA-C*05:01 (uC*05:01). (**B**) Volcano plot displaying relative p8 amino acid frequency from 9mer peptides eluted from 14 HLA-C allotypes based on C1/C2 status. Amino acids with twofold enrichment and statistically significant differences (p<0.05) determined by Student's *t*-test are shown. A total of 26,543 peptide sequences were included. (**C**) Correlation (Pearson) of amino acid volume with amino acid enrichment at p8 of peptides eluted from HLA-C defined by C2/C1 status. Source data available in *Figure 6—source data 1*. Peptide sequences and p8 frequency analysis are available in *Figure 2—source data 2*.

The online version of this article includes the following source data and figure supplement(s) for figure 6:

**Source data 1.** Amino acid frequencies for *Figure 6A*, fold change and p vlaues for 6B, fold change and amino acid volume for 6C.

**Figure supplement 1.** Impact of C1/C2 dimorphism on HLA-C immunopeptidomes.

(pΩ-1) in an analysis of 10mers (*Figure 6—figure supplement 1B*). Next, we analyzed the relative amino acid frequencies at p8 of 9mers eluted from 14 C1 allotypes and 7 C2 allotypes (*Figure 6B*). We found that at p8, C1 allotypes are enriched for Ser, Ala, and Pro, while C2 allotypes are enriched for Phe, Leu, Lys, Met, Arg, Trp, and Tyr (*Figure 6B*, *Figure 6—figure supplement 1C*). There was a strong correlation between amino acid enrichment at p8 in C2 allotypes and amino acid side-chain volume, demonstrating that the C1/C2 dimorphism imposes a size constraint on the peptides bound to HLA-C (*Figure 1C*). In peptides eluted from C1 allotypes, the preference for smaller residues at p8 was accentuated in peptides with pΩ-Ala compared to those with pΩ-Leu (*Figure 6—figure supplement 1D*), suggesting that size at pΩ-1 is especially important for peptides with pΩ-Ala.

## Large amino acid side chains at peptide pΩ-1 position impair TCR9a recognition

Finally, we tested the importance of side-chain size at pΩ-1 by substituting p8 Ser in G12D-9mer and A18L-9mer to Ala, Gly, Glu, Val, Leu, Lys, and Arg. Stimulation of Jurkat-TCR9a by C*08 cells loaded with G12D-9mer p8 substitutions to Leu, Lys, and Arg substantially increased the $EC_{50}$ by approximately 1000-fold, while substitution to Ala, Gly, Glu, Val had a minimal impact on the $EC_{50}$

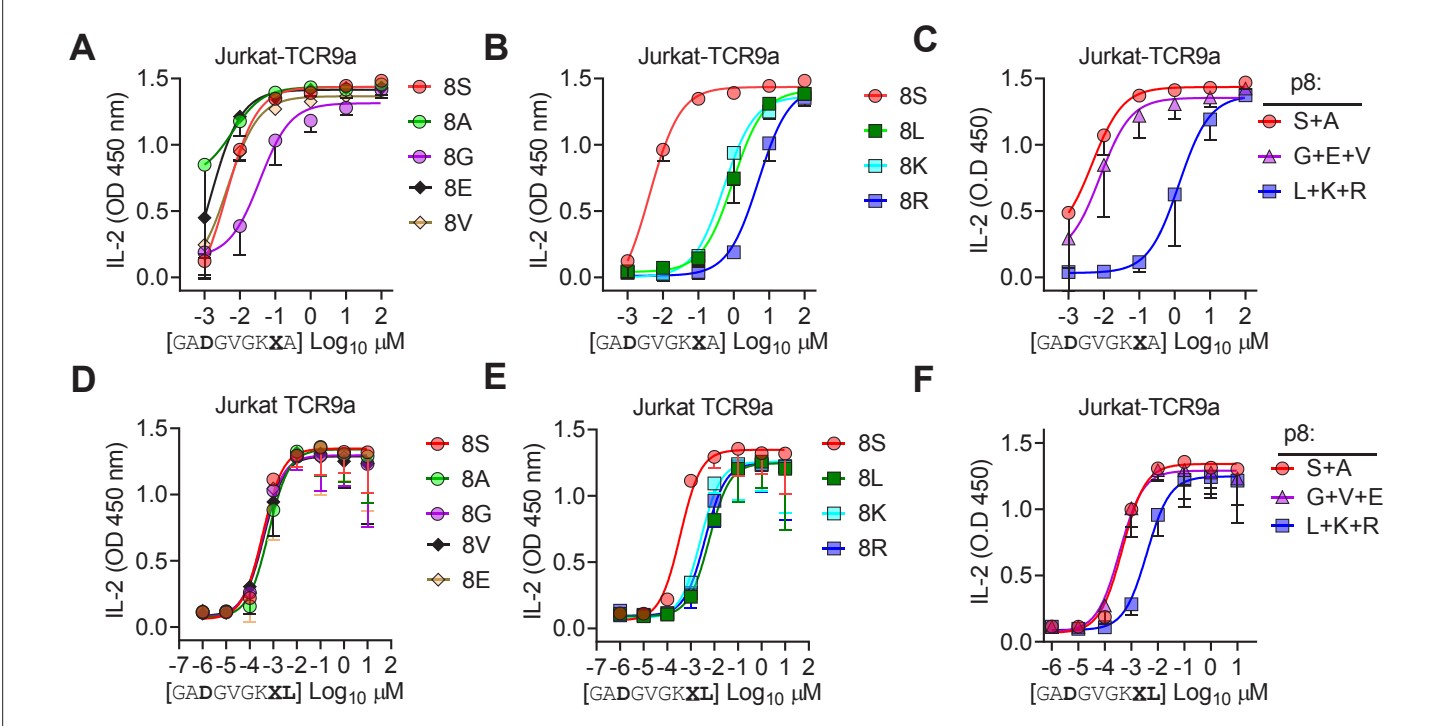

**Figure 7.** Large residues at p8 diminish T cell recognition of C1 HLA-C. (**A–F**) Stimulation of Jurkat-TCR9a⁺ cells by 221C*08:02 cells preloaded with pΩ-1 substitutions of G12D-9mer (**A, B**) and A18L-9mer (**D, E**) at indicated concentrations. Each peptide was tested individually in two independent experiments and displayed by p8 sequence as indicated. (**C, F**) Data from (**A, B**) and (**D, E**) are summarized in (**C**) and (**F**), respectively, by pooling data by indicated p8 substitutions. Means and standard errors of IL-2 concentration in culture supernatant measured by ELISA are shown. Source data available in *Figure 7—source data 1*.

The online version of this article includes the following source data and figure supplement(s) for figure 7:

**Source data 1.** ELISA readings for *Figure 7*.

**Figure supplement 1.** Impact of p8 size on stabilization of HLA-C*08:02.

(*Figure 7A–C*). The same p8 substitutions in A18L-9mer had much less impact on Jurkat-TCR9a stimulation; however, p8 Lys, Arg, and Leu increased the $EC_{50}$ by roughly 10-fold (*Figure 7D–F*). There was not a strict correlation between Jurkat-TCR9a stimulation and stabilization of HLA-C on TAP-deficient C*08⁺ cells (*Figure 7—figure supplement 1*). For example, in G12D-9mer, S8L was a poor ligand for Jurkat-TCR9a, but stabilized HLA-C better than S8G, which was a better ligand for Jurkat-TCR9a. Further, substitution at p8 in A18L-9mer had a minimal effect on HLA-C stabilization, but still impacted Jurkat-TCR9a activation (*Figure 7F*, *Figure 7—figure supplement 1*). This may be due in part to differences in sensitivity between these assays but also suggests a complex relationship between p8 sequence, HLA-C stabilization, and T cell recognition. In line with our other data, the impact of p8 size demonstrates how molecular changes outside TCR contacts can directly impact T cell recognition.

## Discussion

HLA-I molecules form ligands for many immunoreceptors, and the interpretation of disease associations with HLA-I requires modeling interactions with different immune populations (*Debebe et al., 2020*). The C1/C2 dimorphism in HLA-C plays a critical role in innate immunity by forming ligands for the NK cell receptors KIR2DL1 and KIR2DL2/3. By comparing T cell recognition of two HLA-C allotypes identical in sequence except for the C1/C2 dimorphism, we assessed how this dimorphism affects adaptive immunity. We examined two clinically relevant, HLA-C*08:02 restricted neoantigens, G12D-9mer and G12D-10mer derived from the oncogenic hotspot mutation KRAS-G12D. We reject the null hypothesis that the C1/C2 dimorphism exhibits no effect on T cell recognition as C*05 was a poor ligand for TCR9a+ and TCR10+ T cells. Our study identified a clear and underestimated role

for the C1/C2 dimorphism on peptide binding to HLA-C; specifically, as G12D-9mer bound only C*08 and not C*05. Further, we found that the C1/C2 dimorphism shapes the HLA-C immunopeptidome, exerting a size preference at pΩ-1 with C1 allotypes favoring smaller residues. Additionally, we identified an impact of the C1/C2 dimorphism on TCR binding as TCR9a and TCR10 binding to C*05 was weaker than binding to C*08 in the presence of the same peptides.

By investigating the exclusive restriction of TCR9a to C*08, we uncovered a novel role for the C1/C2 dimorphism in peptide presentation. Specifically, the C1/C2 dimorphism defines two distinct PBG structures that impose size constraints at pΩ-1 and allow C1 but not C2 allotypes to bind peptides with shorter pΩ anchors, like Ala in G12D-9mer. The primary cause of these distinct PBGs is likely the size of position 77, which forms a conserved h-bond with the amide of pΩ in both C1 and C2 HLA-C. As Asn (C2) is larger than Ser (C1), it results in peptide pΩ-1 protruding out of the groove, thus permitting larger amino acids at pΩ-1 in C2 allotypes compared to C1. In C1, pΩ-1 sits lower in the groove, restricting pΩ-1 side chains to smaller residues to avoid clashing with HLA-C position Val 76. Consistently, substitutions at p8 in G12D-9mer to the larger Leu, Lys, or Arg substantially reduced C1 recognition by TCR9a, while substitution to Gly had only a minor decrease. Further, the penalty imposed for large residues at p8 was diminished with A18L-9mer, suggesting a canonical pΩ anchor can accommodate unfavorable p8 residues, likely via orientating p8 side chain out of the groove like peptides in C2 allotypes. In C1 allotypes, the additional contacts between Val 76 and p8 may explain the ability of C*08:02 to bind peptides with short, suboptimal pΩ anchors. In contrast, pΩ-1 in C2, which is higher in the PBG and orientated away from the α1 helix, is unable to contact Val 76 and therefore cannot support peptides with pΩ Ala. Consistently, substitution of Asn 77 to Ser in C*05 substantially recovered TCR9a recognition of G12D-9mer.

In addition to peptide binding to HLA-C, TCR-binding affinity and T cell sensitivity were reduced in the presence of C2 residues. Neither N77S or K80N substitutions improved the sensitivity of TCR10 for C*05 and G12D-10mer, suggesting a synergistic effect of the C2 residues on reducing TCR recognition. TCR10 displayed a much slower on rate in binding C*05-G12D-10mer and even after controlling for a poor pΩ anchor, TCR9a was more sensitive to A18L-9mer with C*08 than C*05. Comparing our TCR9a-C*05:01-A18L-9mer structure with that of TCR9a:C*08:02-G12D-9mer revealed no impact on TCR contacts. Further, modeling TCR10 in complex with a recently solved crystal structure of C*05-G12D-10mer revealed no impact on TCR contacts (6JTO) (*Bai et al., 2021*). The most likely explanation for diminished TCR binding in the presence of C2 residues is that of conformational flexibility. Clearly, TCR9a and TCR10 can bind C*05; however, the ability to make the same contacts with C*08 in C*05 likely requires more energetically unfavorable conformational changes, in either peptide, HLA-C, or both, exemplified by the slow on rate of TCR10 with C*05-G12D-10mer. Indeed, recent evidence suggests that TCR recognition can be optimized by peptide substitutions that allow TCR contacts to form more readily (*Devlin et al., 2020*; *Duru et al., 2020*). It is likely that the converse is also true; slight alterations in amino acid side-chain orientations or flexibility may impose a higher energy barrier for efficient TCR recognition.

The G12D-9mer and G12D-10mer are considered public or shared neoantigens due to the high frequency of cancers with KRAS-G12D mutations (*Cox et al., 2014*; *Pearlman et al., 2021*; *Stephen et al., 2014*). Therefore, the small differences between C*08 and C*05 investigated here have clinical implications for mounting an effective antitumor immune response (*Tran et al., 2016*). The ability of C*05 to bind G12D-10mer means HLA-C*05:01-positive individuals may be eligible for such therapies; however, identifying a more sensitive receptor than TCR10 would be needed to translate these findings for clinical application.

The consequences of our findings may have broad implications for T cell recognition of HLA-C beyond the specific example of KRAS-G12D recognition in cancer. For example, in infectious diseases, viral escape mutants may eliminate T cell recognition of C1 allotypes by increasing side chain size at pΩ-1. Similarly, escape mutants that substitute pΩ to smaller residues like Ala may escape T cell recognition of C2 HLA-C but not C1. For peptide vaccines, pΩ-1 size could be optimized depending on whether the peptide targets a C1 or C2 HLA-C allotype. A reasonable prediction from our work is that if a T cell epitope contains a canonical pΩ, it will be immunogenic in individuals carrying either C*08 and C*05. Indeed, there is evidence that HIV-infected individuals can respond to the same epitope (SAEPVPLQL) if they carry HLA-Cw5 or HLA-Cw8 (*Addo et al., 2001*). However, it is possible that preference for pΩ-1 size could engender differential T cell immunity based on differences in HLA-C

and TCR binding. These effects need to be investigated in further experiments of HLA-C-restricted T cells. It is worth noting that the impact of the C1/C2 dimorphism on TCR recognition is intrinsic to the structure of HLA-C and thus may also affect interactions with other immunoreceptors. In particular, the differences in amino acid preferences at pΩ-1 may have enormous consequences for KIR recognition of HLA-C as this residue is central to the KIR binding site (*Boyington et al., 2000*; *Boyington and Sun, 2002*; *Fan et al., 2001*). Our study highlights how small HLA polymorphisms can impact interactions with both peptide and immunoreceptors with implications for immune responses and cancer immunotherapy. Finally, our study establishes that T cells, like NK cells, can discriminate between the dimorphic HLA-C allotypes.

# Materials and methods

## Key resources table

| Reagent type (species) or resource | Designation | Source or reference | Identifiers | Additional information |
|---|---|---|---|---|
| Cell line (human) | 721.221 (221) 221C*05:01 221C*08:02 221C*05:01-N77S 221C*05:01-K80N | Original 221, PMID:3257565 221C*05:01 and 221C*08:02, *Sim et al., 2017*, PMID:28352266 221C*05:01-N77S and 221C*05:01-K80N – this paper | | Plasmids encoding HLA-C*05:01-N77S and HLA-C*05:01-K80N were generated by QuikChange mutagenesis and expressed in 221 cells via retroviral transduction |
| Strain, strain background (*Escherichia coli*) | BL21 (DE3) | Novagen | 70235-4 | For recombinant protein production |
| Recombinant DNA reagent | pet30a-HLA-C*05:01 | This paper | | Synthesized by GenScript, USA Residues 1–278 Sequence from https://www.ebi.ac.uk/ipd/imgt/hla/ Soluble HLA-C*05:01 heavy chain for production in *E. coli* and in vitro refolding |
| Recombinant DNA reagent | pet30a-HLA-C*08:02 | *Sim et al., 2020*; PMID:32461371 | | Soluble HLA-C*08:02 heavy chain for production in *E. coli* and in vitro refolding |
| Recombinant DNA reagent | pet30a-B2M | *Sim et al., 2020*; PMID:32461371 | | Soluble B2M for production in *E. coli* and in vitro refolding |
| Antibody | Anti-HLA-A, -B, -C (mouse monoclonal, clone W6/32) | BioLegend | 311402 | For SPR 10 µg/ml |
| Antibody | Anti-β$_2$m (mouse monoclonal, clone 2M2) APC | BioLegend | 316312 | For peptide-loading experiments 1:50 |
| Antibody | Anti-human CD3 APC-Cy7 (mouse monoclonal) | BioLegend | 300426 | For Jurkat-T cell functional assays 1:100 |
| Antibody | Anti-human CD69 APC (mouse monoclonal) | BioLegend | 555533 | For Jurkat-T cell functional assays 1:100 |
| Cell line (human) | Jurkat-TCR9a, Jurkat TCR10 | *Sim et al., 2020*; PMID:32461371 | | For Jurkat-T cell functional assays |
| Cell line (human) | 221C*05:01-ICP47 | *Sim et al., 2017*; PMID:28352266 | | |
| Cell line (human) | 221C*08:02-TAP-KO | This paper | | Cas9 was expressed in 221C*08:02 cells and gRNA to TAP1 were introduced by lentiviral transduction |
| Recombinant DNA reagent | TCR9a alpha chain | *Sim et al., 2020*; PMID:32461371 | | Soluble TCR9a alpha chain for production in *E. coli* and in vitro refolding |

*Continued on next page*

*Continued*

| Reagent type (species) or resource | Designation | Source or reference | Identifiers | Additional information |
|---|---|---|---|---|
| Recombinant DNA reagent | TCR9a beta chain | *Sim et al., 2020*; PMID:32461371 | | Soluble TCR9a beta chain for production in *E. coli* and in vitro refolding |
| Recombinant DNA reagent | TCR10 alpha chain | *Sim et al., 2020*; PMID:32461371 | | Soluble TCR10 alpha chain for production in *E. coli* and in vitro refolding |
| Recombinant DNA reagent | TCR10 beta chain | *Sim et al., 2020*; PMID:32461371 | | Soluble TCR10 beta chain for production in *E. coli* and in vitro refolding |
| Peptide, recombinant protein | Streptavidin PE | Agilent | PJRS25-1 | For HLA-C tetramers |
| Peptide, recombinant protein | WT and G12D KRAS peptides with substitutions | This paper and *Sim et al., 2020*; PMID:32461371 | | Synthesized by GenScript, USA |
| Commercial assay or kit | Human IL-2 ELISA | BioLegend | 431804 | For Jurkat T-cell functional assays |
| Recombinant DNA reagent | TAP1 gRNA | GenScript | 1 gRNA sequence: CCCAGATGTCTTAGTGCTAC 2 gRNA sequence: ACCTGTAGCACTAAGACATC | pLentiCRISPR v2 vector Used to knockout TAP in 221C*08:02-Cas9 cells by lentiviral transduction |
| Commercial assay or kit | Amine coupling kit | GE Healthcare Life Sciences | BR100050 | For immobilization of protein on SPR chip |
| Commercial assay or kit | CM5 chips | Cytiva | 29149604 | For SPR |
| Peptide, recombinant protein | Biotinylated HLA-C monomers | NIH Tetramer Core Facility | | For HLA-C tetramers |

## Cell lines and culture

Cell lines were cultured in IMDM and 10% fetal calf serum (FCS) at 37°C and 5% $CO_2$. 721.221 (221) cells expressing HLA-C*05:01 and the TAP inhibitor ICP47 (221C*05:01-ICP47) were described previously (*Sim et al., 2017*; *Sim et al., 2019*). Cas9-expressing 221 cells were a generated by lentiviral transduction. cDNA encoding full-length HLA-C*08:02 was expressed in 221-Cas9 cells by retroviral transduction with vectors on the pbabe backbone. Two TAP1 gRNAs (GenScript, USA) were introduced by lentiviral transduction. Jurkat T cells expressing TCR9a and TCR10 were previously described (*Sim et al., 2020*).

## Peptide loading assay

Peptide loading assays were performed as described (*Sim et al., 2017*; *Sim et al., 2019*). 221C*05:01-ICP47 and 221-Cas9-C*08:02-TAP cells were cultured overnight at 26°C with 100 μM of peptide. Cells were then stained with anti-mAb (APC, clone 2M2, BioLegend, USA, #316312). Expression of HLA-C was determined by flow cytometry. Each peptide was tested at least twice in independent experiments. Peptides were synthesized at >95% purity (GenScript).

## T cell activation assay

Peptides were incubated with $10^5$ 221-HLA-C+target cells for 4 hr at 37°C before incubation with $10^5$ Jurkat-TCR+ cells overnight. The following day, cell culture supernatants were recovered, and IL-2 measured by ELISA using the ELISA MAX Deluxe Set Human IL-2, BioLegend (#431804). For assays with TAP-deficient target cells, $10^5$ target cells were incubated overnight at 26°C with peptide ranging from 1 μM to 0.01 nM. The following day, target cells were mixed with TCR+ Jurkat T cells for 6 hr at 37°C. Cells were then washed twice in PBS and stained with mAbs to CD69 (APC, FN50, BD Biosciences, USA, #555533) and CD3 (APC-Cy7, UCHT1, BioLegend, #300426). Expression of CD69 was measured on CD3+ cells by flow cytometry. All peptides were tested at indicated concentrations at least twice in independent experiments.

## HLA-C immunopeptidomics

For the analysis of HLA-C immunopeptidomes, we used published datasets from two studies (*Di Marco et al., 2017*; *Sarkizova et al., 2020*). These data are found in public depositories; Di Marco

et al., PRIDE Project PXD009531, Sarkizova et al., MASSIVE MSV000084172. Both studies used cells transfected with single HLA-I allotypes. Sarkizova et al. used the HLA-A, -B, -C-deficient 721.221 cells, while Di Marco et al. used C1R cells. C1R cells express low levels of HLA-B*35:03 and HLA-C*04:01, and therefore, peptides aligning with motifs for these allotypes were removed from HLA-C allotype-specific sequences. The false discovery rates were 1 and 5% for Sarkizova et al. and Di Marco et al., respectively. For comparing the immunopeptidomes of C1 and C2 HLA-C allotypes, 9mer peptides eluted from 21 different HLA-C allotypes were obtained. The list of allotype-specific sequences from each study was merged and duplicate sequences removed. For some HLA-C allotypes, data was only available from one study and therefore no merging occurred. The C1 HLA-C allotypes were HLA-C*01:02, C*03:02, C*03:03, C*03:04, C*07:01, C*07:02, C*07:04, C*08:01, C*08:02, C*12:02, C*12:03, C*14:02, C*14:03, and C*16:01. The C2 HLA-C allotypes were HLA-C*02:02, C*04:01, C*04:03, C*05:01, C*06:02, C*15:02, and C*17:01. For C*03:02, C*04:03, C*07:04, C*08:01, C*12:02, annd C*14:03, data were only available from one study (*Sarkizova et al., 2020*). A total of 18,275 peptides from C1 allotypes and 8268 peptides from C2 allotypes were studied. The amino acid frequency at positions 8 (pΩ-1) and 9 (pΩ) was determined for each allotype independently and collected to derive an average amino acid frequency for C1 and C2 allotypes. The relative amino acid frequency between C2 and C1 allotypes (C2/C1 fold difference) was determined for each amino acid and significant differences determined by two-sided Student's *t*-test. For comparing C*08 and C*05 immunopeptidomes, a total of 1928 C*05 and 3182 C*08 nonredundant 9mer sequences were compared of which 1196 sequences were shared between C*05 and C*08 (*Figure 5—figure supplement 1*).

## Protein expression, purification, and crystallization

HLA-C*05:01 and TCR proteins were produced largely as described (*Clements et al., 2002*; *Garboczi et al., 1992*; *Sim et al., 2020*; *Tikhonova et al., 2012*). DNA encoding residues 1–278 of HLA-C*05:01, 1–99 of β2M and TCR alpha (1–208) and beta (1–244) chains were synthesized and cloned into the bacterial expression vector pET30a via NdeI and XhoI (GenScript). Proteins were expressed as inclusion bodies in BL21 (DE3) cells (Novagen, USA) and dissolved in 8 M urea, 0.1 M Tris pH 8. Proteins were refolded by rapid dilution in 0.5 M L-arginine, 0.1 M Tris pH 8, 2 mM EDTA, 0.5 mM oxidized glutathione, 5 mM reduced glutathione. Proteins were purified by ion-exchange chromatography (Q column, GE Healthcare, USA) followed by size-exclusion chromatography with a Superdex 200 column (GE Healthcare). HLA-C*05:01-G12D-A18L-9mer and TCR9a were concentrated to 10 mg/ml and crystals grown under the same conditions as TCR9a/d with HLA-C*08:02-G12D-9mer (*Sim et al., 2020*) and were 22% PEG 3350, 0.1 M MOPS pH 7.1, and 0.25 M MgSO4.

## Data collection, structure determination, and refinement

Crystals were immersed in cryoprotectant (crystallization condition plus 20% glycerol) before flash-cooling in liquid nitrogen. A single dataset was collected on the SER-CAT 22 ID beamline (Argonne National Laboratory, IL, USA), processed and merged using HKL2000 (*Otwinowski and Minor, 1997*). TCR9a:HLA-C*05:01-G12D-A18L-9mer structure was solved by molecular replacement method with Phaser in the CCP4 package using the TCR9a:HLA-C*08:02-G12D-9mer complex as a search model (PDB: 6ULR) (*Sim et al., 2020*). The model was built and refined with Coot and Phenix (*Adams et al., 2002*; *Collaborative Computational Project, 1994*; *Emsley and Cowtan, 2004*; *McCoy et al., 2007*; *Otwinowski and Minor, 1997*). The complex contained one complex per asymmetric unit and belonged to the P2₁ space group. Peptide and CDR loops were added manually using 2Fo-Fc electron density maps. Graphical figures were generated in PyMOL.

## Surface plasmon resonance

SPR was performed largely as described (*Sim et al., 2020*) with a BIAcore 3000 instrument and analyzed with BIAevalution software v4.1 (GE Healthcare). The HLA-A, -B, -C-specific mAb W6/32 (#311402, BioLegend) was immobilized to CM5 chips (Cytiva, USA) in 10 mM sodium acetate pH 5.5 at 5000–7000 response units (RU) by primary amine-coupling with a 2 µl/min flow rate. HLA-C was captured by W6/32 at 400–700 RU in PBS. Soluble TCR heterodimers were used as analytes in 10 mM HEPES pH 7.5 and 0.15 M NaCl with a flow rate of 50 µl/min. TCRs were injected for 2 min followed by a dissociation of 10 min. Binding was measured with serial dilutions of TCR from 80 µM to 1.25 µM for TCR10 and 1200 nM to 37.5 nM for TCR9a. Four independent TCR10-binding experiments were

carried out with initial concentrations of 10 µM (two experiments), 40 µM (one experiment), and 80 µM (one experiment). Dissociation constants were obtained by modeling steady-state kinetics for TCR10 and kinetic curve fitting for TCR9a with BIAevaluation software.

## Flow cytometry

Flow cytometry was performed on an LSR II or Fortessa X20 (BD Biosciences) and a Cytoflex S (Beckman Coulter, USA). Data were analyzed using FlowJo software (Treestar V10, USA). Cytometer setup and tracking beads were run daily, and single mAb-stained beads were used to determine compensation settings for multicolor experiments.

## Statistical analysis

Statistical analyses were carried out in GraphPad Prism (version 8) and Microsoft Excel.

## Protein database deposition and files

The C*05:01-A18L-9mer-TCR9a complex was assigned the PDB code: 7SU9. The following PDBs were used in this article: 6ULI, 6ULR, 1EFX, 5VGE, 4NT6, 1QQD, 6JTO, 5VGD, 5W6A, 5W69, and 5W67.

## Acknowledgements

We thank Dr. Sumati Rajagopalan for critical reading of the manuscript. We thank Dr. Ludmila Krymskaya of the LIG Flow Core for assistance with cell sorting and staff at Argonne National Laboratory for assistance with collecting X-ray diffraction data. We thank the NIH Tetramer Core Facility for providing biotinylated HLA-C monomers. This work was supported by the Intramural Research Program of the NIH, National Institute of Allergy and Infectious Diseases.

## Additional information

### Funding

| Funder | Grant reference number | Author |
|---|---|---|
| NIAID Division of Intramural Research Funding | AI000697 | Peter D Sun |
| NIAID Division of Intramural Research Funding | AI000525 | Eric O Long |
| National Institute of Allergy and Infectious Diseases | | Peter D Sun |
| Argonne National Laboratory | | Eric O Long |

The funders had no role in study design, data collection and interpretation, or the decision to submit the work for publication.

### Author contributions

Malcolm J W Sim, Conceptualization, Data curation, Formal analysis, Investigation, Methodology, Project administration, Supervision, Writing – original draft, Writing – review and editing; Zachary Stotz, Paul Brennan, Investigation, Writing – review and editing; Jinghua Lu, Data curation, Formal analysis, Investigation, Methodology, Writing – review and editing; Eric O Long, Conceptualization, Formal analysis, Project administration, Supervision, Writing – original draft, Writing – review and editing; Peter D Sun, Conceptualization, Data curation, Formal analysis, Funding acquisition, Methodology, Project administration, Resources, Supervision, Writing – original draft, Writing – review and editing

### Author ORCIDs

Malcolm J W Sim http://orcid.org/0000-0003-3407-9661

Eric O Long ⓘ http://orcid.org/0000-0002-7793-3728
Peter D Sun ⓘ http://orcid.org/0000-0003-0475-1891

**Decision letter and Author response**
Decision letter https://doi.org/10.7554/eLife.75670.sa1
Author response https://doi.org/10.7554/eLife.75670.sa2

## Additional files

### Supplementary files
• Supplementary file 1. Structural data and refinement statistics for TCR9a-C*05:01-A18L complex. Data for outer shell shown in parentheses.

• Supplementary file 2. Major contacts between TCR9a and HLA-C*08:02 (6ULR) and HLA-C*05:01 (7SU9).

• Transparent reporting form

### Data availability
Diffraction data have been deposited in PDB under the accession code 7SU9.

The following dataset was generated:

| Author(s) | Year | Dataset title | Dataset URL | Database and Identifier |
|---|---|---|---|---|
| Stotz Z, Lu J, Brennan P, Long EO, Sun PD | 2022 | KRAS-G12D specific TCR9a in complex with C*05-GADGVGKSL | https://www.rcsb.org/structure/7SU9 | RCSB Protein Data Bank, 7SU9 |

The following previously published datasets were used:

| Author(s) | Year | Dataset title | Dataset URL | Database and Identifier |
|---|---|---|---|---|
| Sarkizova S, Klaeger S, Oliveira G, Keshishian H, Hartigan CH, Zhang W, Braun DA, Ligon KL, Bachireddy P, Zervantonakis IK, Rosenbluth JM, Ouspenskaia T, Law T, Justeson S, Stevens J, Lane WJ, Eisenhaure T, Zhang GL, Clauser KR, Hacohen N, Carr SA, Wu CJ, Keskin DB | 2020 | Mono-allelic datasets for: A large peptidome dataset improves HLA class I epitope prediction across most of the human population | https://doi.org/doi:10.25345/C5N36Q | MassIVE: Mass Spectrometry Interactive Virtual environment., 10.1038/s41587-019-0322-9 |
| Marco M Di, Schuster H, Backert L, Ghosh M, Rammensee HG, Stevanović S | 2017 | Unveiling the Peptide Motifs of HLA-C and HLA-G from Naturally Presented Peptides and Generation of Binding Prediction Matrices | https://www.ebi.ac.uk/pride/archive/projects/PXD009531 | PRIDE, PXD009531 |

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
