## [Editor Report]

In this study, the authors use structural, functional, and immunopeptidomics analysis to provide insights into how HLA-C C1/C2 dimorphism impacts T cell recognition. This knowledge is important in immunotherapies targeting HLA-C-specific T cells.

---

## [Decision Letter]

**Decision letter after peer review:**

Thank you for submitting your article "T cells discriminate between groups C1 and C2 HLA-C" for consideration by *eLife*. Your article has been reviewed by 3 peer reviewers, and the evaluation has been overseen by a Reviewing Editor and Tadatsugu Taniguchi as the Senior Editor. The following individuals involved in review of your submission have agreed to reveal their identity: Anthony W Purcell (Reviewer #1); Dirk Zajonc (Reviewer #3).

*Reviewer #1 (Recommendations for the authors):*

Points for response buy the authors

1) Was the pHLA complex stability for the WT and heteroclitic peptide assessed for HLA C*05 and C*08??

2) Can the authors comment on the expected conformation of the WT 9mer in C*05 to that observed for the heteroclitic variant? A direct comparison of the bound peptides bound to both C-allotypes would be useful.

3) In Figure 2 the peptidome analysis is shown. In Figure 2G which allotype is the outlier for HLA C1 allotypes? How does this data point impact on the significance? It is important to provide more qualitative information about the data sets used in this analysis = - how do the different data sets compare – some are clearly more comprehensive than others – how many of the datasets are based on monoallelic data vs inferred allotype specificity from other cell lines? Are the two studies performed using similar technology, were the same FDR rates used to report peptides and has this been taken into account in the authors reanalysis? Which databases and search engines were used – it would be most accurate to re-search the raw data together to provide a common analytic workflow to remove biases in the data. At the very least the authors should comment on these influences on the composition of the peptides used for this analysis.

4) It would be interesting to see the results of peptide stabilization assays (akin to T2 stabilization) of the peptides in the TAP deficient 221 cell lines to address the anticipated density of pHLA used for the various TCR readouts/ Likewise was W6/32 flowed over the SPR chis to show similar levels of conformation pHLA were immobilized prior to the SPR analysis? This would seem a critical control?

*Reviewer #2 (Recommendations for the authors):*

1. Supplementary Figure S1B shows that KIR2DL2 binds quite well to 221-C*05:01, would this be a consideration for any therapies targeting the T cell response?

2. Why were 2 different functional readouts utilized for assessing responses from the Jurkat cell lines (e.g. Figure 1C vs Figure S1C)?

3. It was slightly unclear to me why the G12D-A18L peptide was not used for both C*05:01 and C*08:02 for the SPR/tetramer experiments (or as a target for crystallography), given this peptide binds to both alleles and shows improved recognition by TCR9a.

4. The authors state that TCR10 bound C*05 with slightly weaker affinity but this is difficult to judge with such minor differences and large error values.

5. The legend indicates Figure 3C is the summary of 2 experiments and shows mean +/- standard error but the error bars appear to be missing. Moreover, the text indicates the highest concentration of peptide tested was 40 ug/ml, yet only up to 20 ug/ml is shown.

6. The saturation plot in Figure 3F shows a TCR concentration up to 80 μm was used for SPR, yet the highest concentration in 3D and 3E is 40 µM. Was there a reason the highest concentration was not shown on the sensorgrams in Figure 3D and 3E?

7. For Figure 3, it is unclear if the KDs provided are the average KDs from the independent experiments or if it is the KD derived from the curves shown. Was there any variation between the independent experiments or is the large error due to poor curve fitting? Please clarify.

8. It would be good to provide the kinetic fit curves overlaid with the raw data for TCR9a binding to C*05:01-KRAS-G12D-A18L (Figure 3A).

9. What the level of surface expression of the mutant HLA-C*0501-N77S and C*05:01K-80N equivalent to the wild-type?

10. Given the importance of position 77 and the observation that the C*05:01 N77S at least partially recovered recognition by Jurkat TCR9a, is the mutant C*08:01 S77N no longer recognised by TCR9a (or TCR10)? These are important experiments to confirm the importance of position 77.

11. Typo line 53: should be Crohn's disease.

12. Typo line 193: should read Lys (Figure 4C).

13. Typo in figure legend to Sup Figure 1 221-C805:01.

*Reviewer #3 (Recommendations for the authors):*

There is some discussion/speculation about the impact of the different peptides in KIR2DL1-3 recognition. Especially in the scenario where the large amino acid at P(Omega)-1 would diminish T cell recognition. Would the authors think that this would also impact KIR2DL binding and thereby increase NK cell cytotoxicity? Since Dr. Long is an expert in KIR's, these assays should be easy to perform.

---

## [Author Response]

Reviewer #1 (Recommendations for the authors):Points for response buy the authors1) Was the pHLA complex stability for the WT and heteroclitic peptide assessed for HLA C*05 and C*08??

We assessed the ability of KRAS peptides to stabilize C*05 and C*08 using peptide loading assays with TAP-deficient cells expressing C*08 or C*05 (please see Figure 2AandB). WT KRAS peptides were unable to stabilize C*08 or C*05 as they lack a key anchor residue at p3. G12D10mer and A18L-9mer stabilized both C*08 and C*05, while G12D-9mer only stabilized C*08. In addition, G12D-10mer and A18L-9mer successfully refolded recombinant C*05 protein, while in our previous study we found that WT peptides could not refold recombinant C*08 protein (Sim et al., PNAS 2020). Another study also found that WT KRAS peptides and G12D9mer were unable to refold C*05, while G12D-10mer could (Bai et al., Sci China Life, 2021).

2) Can the authors comment on the expected conformation of the WT 9mer in C*05 to that observed for the heteroclitic variant? A direct comparison of the bound peptides bound to both C-allotypes would be useful.

The conformations of WT-9mer and G12D-9mer are hypothetical as neither one can stabilize C*05. We would expect the G12D-9mer conformation in C*05 to be very similar to that of A18L9mer, even though it lacks a suitable C-terminal anchor. Our analysis suggests that while Cterminal sequence is essential for conferring HLA-C stability, it is the interaction between HLA-C position 77 and pΩ-1 side chain size that determines the peptide confirmation near the peptide C-terminus (Figure 5A-J). Indeed, the C-terminal residue conformation was the same when comparing G12D-9mer and G12D-10mer bound to C*08 (Sim et al., PNAS 2020). It is difficult to predict the conformation of WT9mer in C*05 as it lacks both a suitable C-terminal anchor and p3 anchor and would therefore be held on the groove at each end only by the conserved interactions with N and C termini. This would lead to considerable conformational flexibility, and binding would likely be unstable. However, if binding occurred, it is likely that the pΩ-1 conformation would follow the pattern observed in other peptides bound to C2 HLA-C allotypes.

3) In Figure 2 the peptidome analysis is shown. In Figure 2G which allotype is the outlier for HLA C1 allotypes? How does this data point impact on the significance?

The outlier is HLA-C*08:01, and removing the data does not alter the conclusion that C1 allotypes have a higher frequency of p9 Ala. After removing C*08:01, the p-value changes from 0.0098 to 0.0003, by unpaired t test with Welch’s correction.

It is important to provide more qualitative information about the data sets used in this analysis =- how do the different data sets compare – some are clearly more comprehensive than others – how many of the datasets are based on monoallelic data vs inferred allotype specificity from other cell lines? Are the two studies performed using similar technology, were the same FDR rates used to report peptides and has this been taken into account in the authors reanalysis? Which databases and search engines were used – it would be most accurate to re-search the raw data together to provide a common analytic workflow to remove biases in the data. At the very least the authors should comment on these influences on the composition of the peptides used for this analysis.

We appreciate the reviewer's concern that there was not enough detail covering these datasets. We have updated both the corresponding results and methods sections.

We chose these datasets as both studies covered a large number of HLA-C allotypes in the same study, as opposed to pooling studies where only one or two HLA-C allotypes were reported. Our initial analysis used the data by Di Marco et al., as it was published first. After publication of the larger dataset by Sarkizova et al., we analysed this data separately and observed similar amino acid frequencies based on C1/C2 status, reaching the same conclusions. Therefore, when preparing our manuscript, we merged the datasets for a combined analysis, rather than displaying the two analyses separately that confirmed each other. In our original manuscript, these details were omitted from our methods. Furthermore, the reviewer is correct that there are differences between the studies that should also be mentioned including cell line origin, data acquisition and data analysis. Another caveat is that our analysis does not account for peptide abundance.

We have added the following to the corresponding results and methods sections describing our analysis:

Results: ‘To explore if pΩ Ala is a general feature of peptides that bind C1 allotypes (like C*08) and not C2 allotypes (like C*05), we examined C-terminal amino acid usage in the immunopeptidomes of HLA-C allotypes using two publicly available datasets (see methods) (Di Marco et al., 2017; Sarkizova et al., 2020). We initially analysed both datasets independently and observed similar amino acid frequencies, thus we present a combined analysis of a merged dataset with duplicate sequences removed. However, we cannot rule out that differences in cell line origin, data acquisition, false-discovery rate or analytical pipelines may have influenced our analysis. In addition, our analysis was limited to that of amino acid frequency and did not incorporate peptide abundance. We first compared the immunopeptidomes of C*08 and C*05 as…’

Methods: HLA-C immunopeptidomics

For the analysis of HLA-C immunopeptidomes, we used published datasets from two studies (Di Marco et al., 2017; Sarkizova et al., 2020). These data are found in public depositories; Di Marco et al., PRIDE Project PXD009531, Sarkizova et al., MASSIVE MSV000084172. Both studies used cells transfected with single HLA-I allotypes. Sarkizova et al., used the HLA-A, -B, -C deficient 721.221 cells, while Di Marco et al., used C1R cells. C1R cells express low levels of HLA-B*35:03 and HLA-C*04:01 and therefore peptides aligning with motifs for these allotypes were removed from HLA-C allotype specific sequences. The false discovery rate was 1% and 5% for Sarkizova et al., and Di Marco et al., respectively. For comparing the immunopeptidomes of C1 and C2 HLAC allotypes, 9mer peptides eluted from 21 different HLA-C allotypes were obtained. The list of allotype specific sequences from each study were merged and duplicate sequences removed. For some HLA-C allotypes, data was only available from one study and therefore no merging occurred. The C1 HLA-C allotypes were; HLA-C*01:02, C*03:02, C*03:03, C*03:04, C*07:01, C*07:02, C*07:04, C*08:01, C*08:02, C*12:02, C*12:03, C*14:02, C*14:03 and C*16:01. The C2 HLA-C allotypes were; HLA-C*02:02, C*04:01, C*04:03, C*05:01, C*06:02, C*15:02 and C*17:01. For C*03:02, C*04:03, C*07:04 C*08:01, C*12:02, C*14:03, data were only available from one study. A total of 18275 peptides from C1 allotypes and 8268 peptides from C2 allotypes were studied. The amino acid frequency at positions 8 (pΩ-1) and 9 (pΩ) were determined for each allotype independently and collected to derive an average amino acid frequency for C1 and C2 allotypes. The relative amino acid frequency between C2 and C1 allotypes (C2/C1 fold difference) was determined for each amino acid and significant differences determined by two-sided student t-test. For comparing C*08 and C*05 immunopeptidomes, a total of 1928 C*05 and 3182 C*08 non-redundant 9mer sequences were compared of which 1196 sequences were shared between C*05 and C*08 (S Figure 6).’

4) It would be interesting to see the results of peptide stabilization assays (akin to T2 stabilization) of the peptides in the TAP deficient 221 cell lines to address the anticipated density of pHLA used for the various TCR readouts/ Likewise was W6/32 flowed over the SPR chis to show similar levels of conformation pHLA were immobilized prior to the SPR analysis? This would seem a critical control?

We performed peptide loading experiments, the results can be found in Figure 2AandB and S Figure 7. For the SPR experiments, HLA-C molecules were conjugated to CM5 chips via immobilised W6/32. As such, only HLA-C stably bound to W6/32 was analysed for TCR binding and chips were loaded with similar levels of HLA-C.

Reviewer #2 (Recommendations for the authors):1. Supplementary Figure S1B shows that KIR2DL2 binds quite well to 221-C*05:01, would this be a consideration for any therapies targeting the T cell response?

KIR2DL2 binding to C2 HLA-C has been reported before and is dependent on HLA-C bound peptides (Sim et al., 2017, Front. Imm). Blocking KIR antibodies showed some efficacy in preclinical studies, but this is likely through an impact on NK cells (Romagne et al., 2009, Blood). As KIR can be expressed on T cells, it would be wise to deplete inhibitory KIR+ expressing cells prior to adoptive T cell therapy. There is also evidence that KIR expression correlates with PD-1 expression and that blocking both PD-1 and KIR may be beneficial in immunotherapies (He et al., 2018, Drug Des Devel Ther).

2. Why were 2 different functional readouts utilized for assessing responses from the Jurkat cell lines (e.g. Figure 1C vs Figure S1C)?

We used two functional readouts to validate our findings using complementary approaches (flow cytometry and ELISA)

3. It was slightly unclear to me why the G12D-A18L peptide was not used for both C*05:01 and C*08:02 for the SPR/tetramer experiments (or as a target for crystallography), given this peptide binds to both alleles and shows improved recognition by TCR9a.

Our rationale was to prioritise the natural variant (G12D-9mer) as much as possible. This led to the inconsistency identified by the reviewer as we compared A18L-9mer in C*05 with G12D9mer in C*08. However, we believe this inconsistency does not change our conclusion that TCR9a binds HLA-C*08:02 with higher avidity and affinity. For peptide loading experiments, A18L-9mer appeared to stabilise C*05:01 and C*08:02 to similar levels, and was better than G12D-9mer for stabilising C*08:02 (Figure 2AandB). However, G12D-9mer is a good, stable ligand for C*08:02 and in no experiment did C*05:A18L-9mer outperform C*08:02 with G12D-9mer. Hypothetically, if C*05:A18L-9mer was a better ligand for TCR9a than C*08:02-G12D-9mer, then we would agree that the additional experiments with C*08:02-A18L-9mer would be warranted. Our data suggest that peptide stability and C-terminal sequence is not the main source of C*08:02 superiority for TCR9a especially when it also does not appear to explain the superiority of C*08:02 over C*05:01, as a ligand for TCR10.

4. The authors state that TCR10 bound C*05 with slightly weaker affinity but this is difficult to judge with such minor differences and large error values.

The displayed dissociation constants were determined by modelling steady state kinetics and by this analysis, there is only a minor two-fold difference between TCR10 binding to C*08 and C*05. However, by kinetic analysis of association rates, TCR10 had a slower on-rate with C*05:01-G12D-10mer than C*08:02-G12D-10mer. Furthermore, C*05:01-G12D-10mer tetramers had considerably lower avidity for Jurkat-TCR10+ cells than C*08:02-G12D-10mer tetramers. Combining these two results allow us to conclude that the TCR10 interaction with C*05 is weaker than that with C*08 (please see point 7, for more details of TCR10 experiments).

5. The legend indicates Figure 3C is the summary of 2 experiments and shows mean +/- standard error but the error bars appear to be missing. Moreover, the text indicates the highest concentration of peptide tested was 40 ug/ml, yet only up to 20 ug/ml is shown.

The error bars are too small to be observed. The reviewer is correct that the highest concentration tested was 40 ug/ml, however we truncated the x-axis for clarity as it allows the lower tetramer concentrations to be more visible. We include the non-truncated graph for the reviewer.

6. The saturation plot in Figure 3F shows a TCR concentration up to 80 μm was used for SPR, yet the highest concentration in 3D and 3E is 40 uM. Was there a reason the highest concentration was not shown on the sensorgrams in Figure 3D and 3E?

The sensorgrams shown in 3D and 3E were from an experiment where 40µM was the highest TCR concentration. This experiment had less ‘buffer jumps’ than the experiment with an 80µM max TCR concentration, making the sensorgrams more suitable for display (see point 7 for more details of TCR10 experiments). We include the sensorgrams from all four independent experiments for the reviewer.

7. For Figure 3, it is unclear if the KDs provided are the average KDs from the independent experiments or if it is the KD derived from the curves shown. Was there any variation between the independent experiments or is the large error due to poor curve fitting? Please clarify.

For 3A, the dissociation constant for TCR9a with C*05-A18L-9mer (mean and standard deviation) was derived from 12 curves from two independent experiments where association and dissociation rates were fitted separately. To demonstrate good curve fitting, we reanalysed these binding data by fitting association and dissociation simultaneously. This analysis revealed a good fit for all curves, deriving a similar Kd of 214 nM. These curves are now shown in S Figure 3B.

For TCR10 experiments (Figure 3D, E, F) 4 independent experiments were carried out with serial dilutions of TCR10 with different starting TCR concentrations. Our initial two experiments used a highest concentration of 10 µM, however this was below the KD of TCR10 for C*05-G12D10mer and therefore to ensure binding saturation, we repeated TCR10 binding with an additional two experiments. The first used a maximum concentration of 40 µM and the second experiment a highest of 80 µM. We reanalysed all binding curves for TCR10 binding experiments to recalculate the dissociation constants. Combining binding data from all four experiments to derive a single dissociation constant gave Kd=7 µM and Kd=14 µM for C*08 and C*05 respectively, but with large confidence intervals. Deriving constants from each binding experiment separately and taking a mean and standard deviation, gave dissociation constants of Kd=8±5 µM and 13±6 µM for C*08 and C*05 respectively. Thus, by both analysis methods TCR10 had weaker binding to C*05 than C*08 but with large overlap in confidence/standard deviation. The variation does not appear to due to poor fitting but due to one experiment (3) where the Kd was higher for both C*08 and C*05. In three of the four experiments, TCR10 binding to C*08 was stronger than with C*05. In the remaining experiment, binding was similar. We include all binding curves for the reviewer.

We have modified the legend to read

“(F) Mean, standard deviation and non-linear curve fitting of TCR10 binding to G12D-10mer bound to C*08:02 and C*05:01. Dissociation constants were derived from four independent binding experiments with two-fold serial dilutions of TCR10 starting at 10 µM (two experiments), 40 µM (one experiment) and 80 µM (one experiment).”

8. It would be good to provide the kinetic fit curves overlaid with the raw data for TCR9a binding to C*05:01-KRAS-G12D-A18L (Figure 3A).

We include fitted curves for both experiments in S Figure 3A.

9. What the level of surface expression of the mutant HLA-C*0501-N77S and C*05:01K-80N equivalent to the wild-type?

We apologize for not including this data, the expression levels of the mutant cell lines were similar to that of WT C*08 and C*05. This data is now shown in new S Figure 5.

10. Given the importance of position 77 and the observation that the C*05:01 N77S at least partially recovered recognition by Jurkat TCR9a, is the mutant C*08:01 S77N no longer recognised by TCR9a (or TCR10)? These are important experiments to confirm the importance of position 77.

These data are shown in the same figure (Figure 5K). As C*05:01 and C*08:02 differ only by positions 77 and 80, C*08:02 – S77N (77N 80N), is the same as C*05:01 K80N (77N 80N), While the N77S substitution in C*05:01 (S77 K80) was able to substantially recover TCR9a activation, it had no impact on TCR10.

11. Typo line 53: should be Crohn's disease.

We have corrected the typo.

12. Typo line 193: should read Lys (Figure 4C).

We have corrected the typo.

13. Typo in figure legend to Sup Figure 1 221-C805:01.

We have corrected the typo.

Reviewer #3 (Recommendations for the authors):There is some discussion/speculation about the impact of the different peptides in KIR2DL1-3 recognition. Especially in the scenario where the large amino acid at P(Omega)-1 would diminish T cell recognition. Would the authors think that this would also impact KIR2DL binding and thereby increase NK cell cytotoxicity? Since Dr. Long is an expert in KIR's, these assays should be easy to perform.

We thank the reviewer for their evaluation of our paper and for this question. We have studied the impact of peptide sequence on KIR recognition in published works (Sim, MJW et al., 2017, Frontiers in Immunology and Sim, MJW et al., 2019, PNAS) and in as yet unpublished data. We can confirm that large residues at p8 are not tolerated by KIR2DL2/3 binding to C*08, while they are by KIR2DL1 binding to C*05. These data are part of an extensive study of peptide sequence recognition by different KIRs and thus are not presented in this publication. We include some unpublished data demonstrating the impact of side chain size on inhibitory KIR binding for the reviewer.

References

Apps R, et al. Influence of HLA-C expression level on HIV control. Science. 2013 Apr

5;340(6128):87-91. doi: 10.1126/science.1232685. PMID: 23559252; PMCID: PMC3784322.

Bai, P., Zhou, Q., Wei, P. *et al.* Rational discovery of a cancer neoepitope harboring the KRAS G12D driver mutation. *Sci. China Life Sci.* 64, 2144–2152 (2021).

https://doi.org/10.1007/s11427-020-1888-1

He Y, Liu S, Mattei J, Bunn PA Jr, Zhou C, Chan D. The combination of anti-KIR monoclonal antibodies with anti-PD-1/PD-L1 monoclonal antibodies could be a critical breakthrough in overcoming tumor immune escape in NSCLC. Drug Des Devel Ther. 2018 Apr 24;12:981-986.

doi: 10.2147/DDDT.S163304. PMID: 29731605; PMCID: PMC5923225.

Romagne F, Andre P, Spee P, Zahn S, Anfossi N, Gauthier L, et al. Preclinical characterization of 1-7F9, a novel human anti-KIR receptor therapeutic antibody that augments natural killermediated killing of tumor cells. Blood. 2009;114(13):2667-77.

Sim, M.J.W., Lu, J., Spencer, M., Hopkins, F., Tran, E., Rosenberg, S.A., Long, E.O., and Sun, P.D. (2020). High-affinity oligoclonal TCRs define effective adoptive T cell therapy targeting mutant KRAS-G12D. Proc Natl Acad Sci U S A.

Sim MJW, Rajagopalan S, Altmann DM, Boyton RJ, Sun PD, Long EO. Human NK cell receptor KIR2DS4 detects a conserved bacterial epitope presented by HLA-C. Proc Natl Acad Sci U S A. 2019;116(26):12964-73.

Sim MJ, Malaker SA, Khan A, Stowell JM, Shabanowitz J, Peterson ME, et al. Canonical and

Cross-reactive Binding of NK Cell Inhibitory Receptors to HLA-C Allotypes Is Dictated by Peptides Bound to HLA-C. Front Immunol. 2017;8:193